# Optimal Anisotropic Guided Filtering in retinal fundus imaging: A dual approach to enhancement and segmentation

**G. Tirumala Vasu**[1], **Samreen Fiza**[2], **Subba Rao Polamuri**[3], **K. Reddy Madhavi**[4], **Thejaswini R**[5], **Venkataramana Guntreddi**[6]*

1 Innovation & Translational Research Hub, Department of Electronics & Communications Engineering, Presidency University, Bangalore, Karnataka, India, 2 Department of Electronics & Communications Engineering, Presidency University, Bangalore, Karnataka, India, 3 Department of Computer Science and Engineering, Aditya University, Surampalem, Andhra Pradesh, India, 4 Department of AI & ML, School of Computing, Mohan Babu University, Tirupati, Andhra Pradesh, India, 5 Department of Electronics & Communications Engineering, Sri Siddhartha School of Engineering, Tumkur, Karnataka, India, 6 Department of Electrical, Telecommunication and Computer Engineering, Kampala International University-Western Campus, Bushenyi, Uganda

* gvramana@kiu.ac.ug

## Abstract

Retinal vascular tree segmentation and enhancement has significant medical imaging benefits because, unlike any other human organ, the retina allows non-invasive observation of blood microcirculation, making it ideal for the detection of systemic diseases. Many traditional methods of segmentation and enhancement encounter issues with visual distortion, ghost artifacts, spatially inconsistent structures, and edge information preservation as a result of the diffusion of spatial intensities at the edges. This article introduces an Optimal Anisotropic Guided Filtering (OAGF) framework tailored for retinal fundus imaging, addressing both enhancement and segmentation needs in a unified approach. The proposed methodology consists of three stages, in the first stage, we perform the illumination correction and then convert the source RGB image to YCbCr format. The luminance (Y) component is further processed through OAGF. In the second stage, optimized top-hat transform and homomorphic filtering has been performed to get segmented image. In the third stage, the enhanced image is produced by converting YCbCr to RGB format. To validate the effectiveness of the suggested approach, extensive experiments with the open-source DRIVE and STARE datasets were performed. Quantitative and qualitative assessments prove that the OAGF-enhancement and segmentation methodology surpasses current algorithms with better values in Dice Coefficient (0.860, 0.854), Precision (0.845, 0.834), and F1 Score (0.827, 0.817) on both databases.

**Data availability statement:** https://figshare.com/s/90317d14f5c06e69015b.

**Funding:** The author(s) received no specific funding for this work.

## 1. Introduction

Helmholtz [1] made it possible to acquire retinal images by inventing the ophthalmoscope, which established ophthalmology as a distinct branch of medicine. The scientific focus on separating retinal blood vessels from fundus images in recent years has piqued researcher's interest in identifying a variety of illnesses, including senile maculopathy, glaucoma, retinal thrombosis, among others. Furthermore, retinal images can be used to diagnose neurological conditions including stroke and cognitive dysfunction [2].

It is evident that the diagnostic capabilities of an image are enhanced by the increased clarity of the image, which allows for the observation of more detailed information. U.K. BioBank conducted a screening study that revealed approximately 30% of fundus images are inadequate for clinical diagnosis [3] or automatic fundus image analysis applications such as optic disc/cup detection [4], retinal vessel segmentation [5], diabetic retinopathy grading [6]. Hence segmentation and enhancement of retinal fundus images becomes a necessity due to above reasons.

Traditional methods and deep learning technology-based methods are two categories into which a variety of enhancement methods for retinal images have been proposed. Traditional approaches consist of Retinex theory-based [7], dark channel prior [8], filter-based [9] and histogram equalization [10,11]. The dynamic range of pixels is stretched to modify image contrast through Histogram Equalization (HE) methods. The Red, Green, and Blue (RGB) channels are enhanced by a luminance gain matrix that is generated by gamma correction of the value channel in the Hue, Saturation, and Value (HSV) colour space.

CLAHE (Contrast-Limited Adaptive Histogram Equalization) [12] is then employed to improve contrast in the luminosity channel. A number of decomposition-based methods [13,14] have also been created, alongside HE-based methods. A reflectance map and an illumination map are the two main components of an image, according to Retinex theory. By modifying the distribution of the illumination map, Retinex-based approaches improve the image. By using a structural prior to modify the image illumination and computing the maximum value of the RGB channels independently to estimate the pixel brightness, Guo et al. [15] enhanced low-light image quality.

A significant number of learning-based approaches [16] rely on fully supervised learning, which necessitates training with carefully aligned image pairs. By modelling the elements that affect fundus images during ophthalmoscope imaging such as artifacts caused by blurred images, light transmission disturbances, and image blurring, a model [17] for fundus image deterioration was proposed. Unsupervised approaches [18] have also been developed alongside supervised methods.

In order to guarantee that fundus images are adequately illuminated and structurally stable during training, Ma et al. [19] enhanced CycleGAN [20] with illumination and structure constraints. Semi-supervised techniques are proposed to leverage the benefits of both supervised and unsupervised learning in scenarios with limited labels. By including the total variation prior with the dark channel prior, a semi-supervised model [21] for image dehazing was proposed. Directly applying

these methods to fundus image enhancement may not yield desirable outcomes because they typically train models on task-specific prior knowledge of natural scenes.

Methods for medical image segmentation have been the subject of substantial research over the last several decades. Using morphological reconstruction and centerline extraction to identify blood arteries at the local level focusing on a single vessel and disregarding their global relationships, a retinal image segmentation approach [22] was proposed. Despite the fact that graph-based methods disseminated global information by means of topological map [23] construction, conventional image processing-based methods required manual pruning [24] of numerous hyperparameters [24], which could lead to an increase in computational complexity. The vascular tree present in the retinal fundus images can be improved using a Hessian-based multiscale filter [25].

The filter examines the tubular shape by the second-order derivatives of the vessels to classify the pixels as part of the vessels. Aortic structures in fundus images were extracted using an enhanced straight-line detector and a hidden Markov model [26] was used to identify the vascular centreline and the fine blood vessels. The optic disc and optic cup were segmented using super-pixel classification [27] on the basis of pixel values. Some extensive surveys like that by Khan et al. [28] have discussed the difficulties, classification paradigms, and current developments in retinal blood vessel extraction methods.

Traditional and deep learning-based enhancement and segmentation techniques frequently fight to optimize noise removal and maintain vessel structure integrity. Diffusion methods are prone to blur tiny details and learning methods need huge amounts of diverse data and tend to generalize less well under poor imaging conditions. Hence, there is an acute need for an approach to enhance the quality of an image and segment fine vascular structures well under poor conditions. A hybrid model combining region-based enhancement with unsupervised segmentation, showing promise in preserving vascular structures without requiring ground-truth labels [29].

To provide a clearer understanding of the limitations of existing segmentation techniques, we summarize the key shortcomings of both traditional and deep learning-based methods in Table 1.

To solve this, we introduce an Optimal Anisotropic Guided Filtering (OAGF) framework. In this approach, enhancement and segmentation are to be performed in one pass by utilizing structure adaptivity, direction smoothing and contrast-preserving filtering. The OAGF framework is intended to prevent spatial information loss and maintain microstructure of blood vessels.

The remainder of this article is organized as follows: Section 2 introduces related work and expounds on the framework of Optimal Anisotropic Guided Filtering (OAGF). Section 3 presents the proposed approach to retinal image enhancement and segmentation. Section 4 showcases experimental findings and qualitative and quantitative comparisons on publicly available datasets. Finally, in Section 5, the article concludes by summarizing the results and some directions for future work.

## 2. Related work on Optimal Anisotropic Guided Filter (OAGF)

The application of anisotropic diffusion to the images leads to the generation of a smooth texture in uniform regions, while effectively maintaining the fidelity of the sharp edges and varied sections of the blurred image. A Bilateral Filter (BF) in

**Table 1. Summary of limitations in traditional vs. deep learning-based segmentation methods.**

| Criteria | Traditional Methods | Deep Learning Methods |
|---|---|---|
| Parameter Dependency | Require manual tuning of thresholds and operators | Sensitive to hyperparameter and architecture choices |
| Data Requirement | Perform adequately on small datasets | Require large annotated datasets for effective training |
| Robustness to Noise | Struggle in noisy or low-contrast fundus images | Performance degrades under poor imaging conditions |
| Generalizability | Limited adaptability across patient variability | Susceptible to overfitting and dataset bias |
| Interpretability | Transparent and explainable but less powerful | Often black-box, limiting clinical interpretability |
| Computational Efficiency | Low computational burden, real-time capable | High resource demand for training and inference |

image enhancement and segmentation is a non-linear filtering technique that effectively smooths an image while simultaneously preserving sharp edges. The BF tends to blur small-scale features or textures because it prioritizes preserving stronger edges. Fine details, such as delicate textures or subtle features, may be lost or overly smoothed.

Anisotropic Diffusion (AD) is essential for maintaining edges with a sharpness similar to the original image. Depending on the implementation and choice of the gradient operator, AD can introduce directional artifacts, especially in regions with high anisotropy or irregular structures. These artifacts can detract from the natural appearance of the enhanced image. The Guided Filter (GF) uses a guidance image to control the smoothing process, making it edge-aware and avoiding halos. The problem with GF is, it will assume a linear model in local patches and struggle with very fine or complex edge structures. The Anisotropic Guided Filter (AGF) achieves a balance between complexity and flexibility, rendering it exceptionally efficient for particular applications. Anisotropic behaviour focuses on directional smoothing, which can sometimes introduce directional artifacts if the weights are improperly designed or tuned. These artifacts may be visually noticeable in regions with highly irregular or isotropic structures. Table 2 presents a key comparison of different edge preserving filters.

The proposed Optimal Anisotropic Guided Filter (OAGF) improves upon the AGF by introducing optimal behaviour, allowing directional smoothing and better edge preservation. It modifies the traditional AGF by incorporating optimal weights to handle varying edge orientations in an image more flexibly and implement an approximate region-selective diffusion procedure.

Equation (1) represents the recommended trade-off between edge preservation and smoothing, as put out by Perona and Malik [34], where the image is smoothed iteratively based on the local gradient.

$$I'_{(x,y)} = \nabla.[DCM\left(\left\|\nabla I^t_{(x,y)}\right\|\right)\nabla I^t_{(x,y)}]$$

(1)

where, $t$ denotes the time parameter, $I^0_{(x,y)}$ indicates the original image, $\nabla I^t_{(x,y)}$ signifies the gradient function $I_{(x,y)}$ and $DCM(\cdot)$ corresponds to the Diffused Conductance Model that controls how much diffusion (smoothing) occurs at each pixel. In high-gradient areas (edges), the conductance is low to preserve detail, while in flat regions, higher diffusion smooths out noise. This enables edge-preserving filtering, which is essential in medical images where vessel boundaries must be maintained.

The proposed OAGF is contingent upon the quantity of the spatial variable influenced by diffusion at the boundaries. To enhance the performance of edge detection in diffused spatial regions, the image structural information [35] is being considered as given in Equation (2). The sharpness characteristics of an image are primarily conveyed through its gradient's parameter. An over blurred image exhibits a peculiar quality resulting from a decrease in gradient information. Focused areas have more gradient information than defocused ones, and it stands out more. Consider a discreate image from $I_{(x,y)}$ as $I_Z$. The eight gradients of Structural Patch Elements ($STP$) for the $I_Z$ are given in Equation (2).

$$\nabla_N STP_{(x,y)} = [STP]_{(x-1,y)} - [STP]_{(x,y)}$$

$$\nabla_S STP_{(x,y)} = [STP]_{(x+1,y)} - [STP]_{(x,y)}$$

**Table 2. Performance analysis of different edge preserving filters.**

| Filter | Edge-Preserving Ability | Computational Efficiency | Artifacts | Adaptability to Structures |
|---|---|---|---|---|
| BF [30] | Moderate | Low | Halo artifacts | Limited |
| GF [31] | Good | High | Minimal halos | Moderate |
| AD [32] | Excellent | Low (iterative) | Few artifacts | High |
| AGF [33] | Excellent | Moderate to High | Minimal halos | Very High |

$$\nabla_E STP_{(x,y)} = [STP]_{(x,y+1)} - [STP]_{(x,y)}$$

$$\nabla_W STP_{(x,y)} = [STP]_{(x,y-1)} - [STP]_{(x,y)}$$

$$\nabla_{NE} STP_{(x,y)} = [STP]_{(x-1,y+1)} - [STP]_{(x,y)}$$

$$\nabla_{NW} STP_{(x,y)} = [STP]_{(x-1,y-1)} - [STP]_{(x,y)}$$

$$\nabla_{SE} STP_{(x,y)} = [STP]_{(x+1,y+1)} - [STP]_{(x,y)}$$

$$\nabla_{SW} STP_{(x,y)} = [STP]_{(x+1,y-1)} - [STP]_{(x,y)} \qquad (2)$$

where $\nabla_N$, $\nabla_S$, $\nabla_E$, $\nabla_W$, $\nabla_{NE}$, $\nabla_{NW}$, $\nabla_{SE}$, and $\nabla_{SW}$ are diffusion gradients with intensity variation in the direction of north, south, east, west, north-east, north-west, south-east and south-west respectively. $STP = \sum_{i=1}^{N} STP_i \quad \forall \ STP_i \in I'_{(x,y)}$. In this context, $STP(\cdot)$ represents the fundamental structural patch element with a radius of '$r$', the total number of scales is denoted as '$N$', and $STP_i$ signifies the $i^{th}$ layer of the structural patch element. Rather than employing one gradient alone, we examine intensity differences in eight directions: north, south, east, west, and diagonals. Patch-level representation of gradients makes the model sensitive to local edge orientations so more precise detection is facilitated in thin and curved vessels' structures. It imitates clinicians' visual evaluation of vascular patterns from several perspectives.

Edges and textures must be precisely maintained in images. In the first iteration, the image's texture is more noticeable. The image becomes smoother after the second and third iterations, which is not necessary for image upscaling. Therefore, only the first iteration is specified. More blurriness is seen in the image's outermost areas as the iteration continues, while finer features become less apparent. The aim is to extract the boundary as well as small local features.

The Coefficient of Diffusion($DC(\cdot)$), dependent on the gradient or discontinuity of the image, can be expressed as the norm of the gradient, as outlined in Equation (3).

$$DC_{N(x,y)} = I_{(x,y)}^{t+1} - I_{(x,y)}^{t} * \nabla_N STP_{(x,y)}$$

$$DC_{S(x,y)} = I_{(x,y)}^{t+1} - I_{(x,y)}^{t} * \nabla_S STP_{(x,y)}$$

$$DC_{E(x,y)} = I_{(x,y)}^{t+1} - I_{(x,y)}^{t} * \nabla_E STP_{(x,y)}$$

$$DC_{W(x,y)} = I_{(x,y)}^{t+1} - I_{(x,y)}^{t} * \nabla_W STP_{(x,y)}$$

$$DC_{NE(x,y)} = I_{(x,y)}^{t+1} - I_{(x,y)}^{t} * \nabla_{NE} STP_{(x,y)}$$

$$DC_{NW(x,y)} = I_{(x,y)}^{t+1} - I_{(x,y)}^{t} * \nabla_{NW} STP_{(x,y)}$$

$$DC_{SE(x,y)} = I_{(x,y)}^{t+1} - I_{(x,y)}^{t} * \nabla_{SE} STP_{(x,y)}$$

$$DC_{SW(x,y)} = I_{(x,y)}^{t+1} - I_{(x,y)}^{t} * \nabla_{SW} STP_{(x,y)} \qquad (3)$$

The response of the proposed OAGF will consider the Equation (3) gradients and the linear model of OAGF is given in Equation (4).

$$OAGF_i = \sum_{k|i \in STP} STP(\cdot)\left(a_k\,DC(\cdot) + b_k\right)$$

(4)

The solutions $a_k$ and $b_k$ are obtained by solving modified minimization problem as described by Equation (5).

$$a_k = \frac{\sum_{i \in STP(\cdot)} STP(\cdot)DC(\cdot)\left\{I_i^{t+1} - I_i^t\right\}}{\sum_{i \in STP(\cdot)} STP(\cdot)\left(DC(\cdot) - \mu_k\right)^2 + \epsilon d_x}$$

$$b_k = I_i^t - a_k\mu_k$$

(5)

where, $\mu_k$ is mean of eight gradients. Three parameters highly influence the OAGF, namely, stopping time of iteration process $'t'$, the directional (north, south, east, west, north-east, north-west, south-east and south-west) regularization $'d_x'$ and regularization parameter $'\epsilon'$ which determines the smoothness of $a_k$. OAGF introduces a modified optimization strategy where the coefficients $a_k$ and $b_k$ are dynamically derived using a gradient-weighted approach, enabling region-aware and directionally adaptive smoothing. This improves structural preservation in fundus images compared to conventional filters.

The proposed methodology, explained in the next chapter, will incorporate further utilization of the OAGF and its edge-preserving properties in applications of segmentation and enhancement.

## 2.1. Distinctive features of OAGF vs. existing edge-preserving filters

### 2.1.1. Mathematical advancements.
Traditional guided filters, like GF, make a local linear assumption and may not perform well with fine or complicated edge structures. Although the anisotropic behaviour is added in AGF, it still has no region-specific optimization for gradients. The limitation of this is addressed by the presented OAGF, in which an optimal conductance mechanism includes local structural knowledge.

As shown in Equation (2), OAGF introduces multi-directional gradient patch elements capturing intensity variations across eight directions North, South, East, West, and diagonals enhancing sensitivity to vessel curvature and orientation. A modified diffusion conductance function, as shown in Equation (3), which considers the norm of spatial gradients to guide filtering strength in edge-preserving fashion. An adaptive optimization framework as given in Equation (5) where the coefficients controlling the filter (α, β) are dynamically updated using a weighted gradient strategy, thus improving context-aware smoothing and structural fidelity.

### 2.1.2. Structural awareness and regional adaptivity.
Unlike GF or BF, which apply spatially uniform or fixed-range filters, OAGF is region-adaptive, using localized structure-aware weighting functions that dynamically adjust filtering strength based on underlying anatomical cues. The use of structural patch elements ensures that vessel boundaries especially microvascular bifurcations and curved vessels are retained with high fidelity. This is especially true in retinal fundus images, where subtle clinical signs such as microaneurysms and capillary loops are easily masked by traditional filters. The fact that OAGF is responsive to local anatomical geometry ensures maximum detail enhancement without losing important information for diagnosis.

OAGF is a substantial improvement over current edge-preserving filter. It provides a novel combination of directional anisotropy, gradient-weighted controls over diffusion, patch-based structural adaptivity and combined support for both pixel-wise image enhancement and segmentation. These functions make OAGF especially well-suited for medical imaging when precision and structural preservation are of utmost concern. Further, Table 3 shows the practical distinctions between the proposed OAGF and the conventional AGF.

**Table 3. Comparison between AGF and OAGF in practical deployment terms.**

| Feature/ Criterion | AGF | Proposed OAGF |
|---|---|---|
| Application Integration | Commonly used for Enhancement only | Unified pipeline for enhancement and segmentation |
| Edge Awareness | Moderately sensitive; some directional artifacts | Highly adaptive; optimized via gradient magnitude and local structure |
| Parameter tuning | Fixed or empirically tuned | Dynamically optimized coefficients based on weighted gradient measures |
| Structural Patch | Not structurally aware | Yes, region-adaptive patch design to preserve vessel bifurcations and curves |
| Conductance Function | Static or heuristically defined | Optimally computed from normalized gradient norm for context-aware filtering |
| Iteration Control | Not explicitly optimized | Optimally tuned stopping time for preserving microvascular structures |
| Clinical Utility (Retinal Fundus) | Moderate; prone to noise or vessel blurring | High; superior vessel continuity and edge fidelity across varying conditions |

## 3. Retinal image segmentation and enhancement using proposed OAGF

Fig 1 shows the block diagram that represents a comprehensive framework for segmentation and enhancement of retinal image, ensuring improved visibility of retinal features and robust segmentation of critical structures. The method is divided into three stages: illumination correction and OAGF filtering, segmentation via optimized top-hat transform and homomorphic filtering, and enhanced image reconstruction.

### 3.1. Stage 1: Illumination correction and OAGF filtering

As shown in Fig 1, the first stage addresses uneven illumination, which is a common challenge in fundus imaging. Illumination variations arise due to optical and hardware constraints during image acquisition. This stage employs several steps to ensure uniform illumination and enhanced contrast for better feature representation. A retinal fundus image $\left(I^S_{(x,y)}\right)$ incorporating essential diagnostic elements, including the optic disc, blood vessels, and several retinal structures is the input to this framework. Due to lighting inconsistencies these features are often obscured.

Content-aware filtering enhances the local contrast of the image based on spatial features. This dynamic adjustment [36,37] ensures that the enhancement is focused on diagnostically relevant areas while avoiding over-amplification of noise. The filtering process is mathematically described in Equation (6).

$$W(x, y, u, v) = e^{\left(\frac{\left|I^S_{(x,y)} - I^S_{(x+u,y+v)}\right|^2}{2\sigma^2_{si}}\right)} . e^{\left(-\frac{u^2+v^2}{\sigma^2_{sn}}\right)}$$

$$I^{CAF}_{(x,y)} = \frac{\sum_{u=-r}^{r} \sum_{v=-r}^{r} W(x, y, u, v).H(u, v).I^S_{(x+u,y+v)}}{\sum_{u=-r}^{r} \sum_{v=-r}^{r} W(x, y, u, v).H(u, v)}$$

(6)

where, $H(u, v)$ is a spatial kernel of radius '$r$' applied around pixel $(x, y)$, $W(x, y, u, v)$ is a weight function dependent on intensity similarity and spatial distance, $\sigma_{si}$ controls the sensitivity to intensity differences, and $\sigma_{sn}$ controls the spatial influence of neighbouring pixels.

Equation (6) penalizes large intensity differences between a central pixel and its neighbours ensuring that pixels with similar brightness values contribute more to the output. This helps preserve edges and vessel boundaries, which typically have strong intensity contrasts. And pixels that are farther from the centre have lower influence. This creates a local context window, focusing the filter's operation on a small neighbourhood around each pixel.

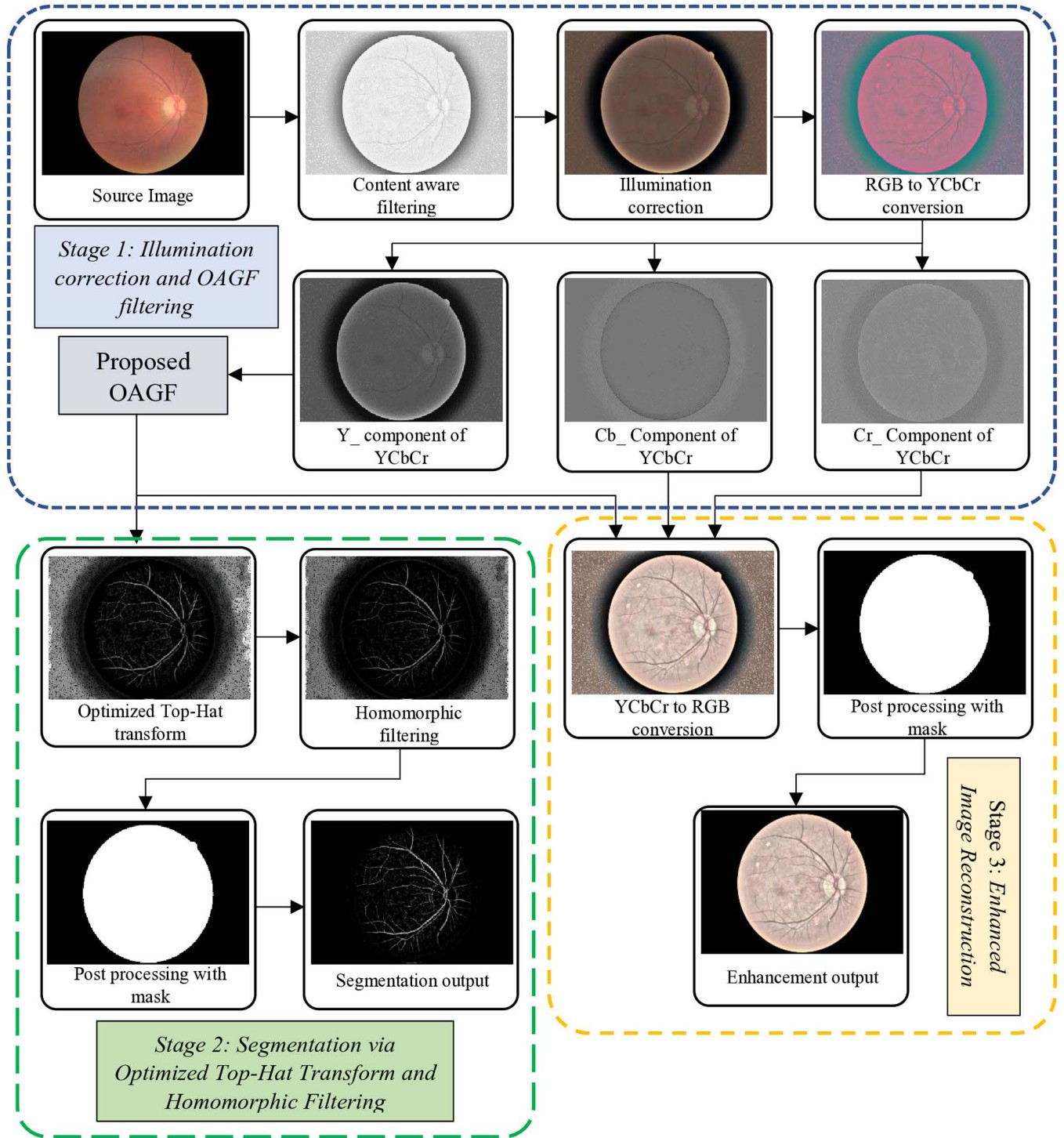

**Fig 1. Block diagram of retinal image segmentation and enhancement using proposed OAGF.**

Illumination inconsistencies are corrected using homomorphic filtering. This technique separates illumination and reflectance components of the image by applying a high-pass filter as given in Equation (7).

$$I^{ill}_{(x,y)} = HPF\left(I^{CAF}_{(x,y)}\right)$$ (7)

As shown in Fig 1, RGB to YCbCr conversion has to be performed (given in Equation (8)) to separate the luminance (Y) (brightness information) from the chrominance (Cb and Cr) (colour information). In the RGB colour space, brightness and colour are intertwined. Any operation on brightness (e.g., contrast adjustment) can unintentionally alter colour balance. Converting to YCbCr separates brightness (Y) from colour information, making contrast enhancement easier and more reliable.

$$RGB2YCbCr\left(I^{ill}_{(x,y)}\right) = [I^{Y}_{(x,y)},\ I^{Cb}_{(x,y)}, I^{Cr}_{(x,y)}]$$ (8)

The luminance $((I^{Y}_{(x,y)})$ channel, which carries most of the intensity information, is further processed by proposed OAGF as given in Equation (9). The filter kernel adapts dynamically based on local intensity gradients, ensuring that both global and local features are preserved. The detailed explanation of implementation of OAGF is given in chapter 2.

$$O\hat{A}GF = OAGF^{Y}_{(t,d_x,\epsilon)} = OAGF(I^{Y}_{(x,y)})$$ (9)

By the end of this stage, the framework generates an illumination-corrected image with enhanced contrast and feature clarity, ready for segmentation and enhancement.

### 3.2. Stage 2: Segmentation via optimized Top-Hat transform and homomorphic filtering

This stage focuses on extracting key diagnostic features, particularly the vascular structures, which are essential for retinal image analysis through optimized Top-Hat transform and homomorphic filtering. When applied to a grayscale 2-D image of the eye's fundus, mathematical morphology which consists of algebraic arithmetic operators. Becomes a potent tool for image processing.

When comparing an input image with its morphologically closed-form, the Top-Hat morphological operation [38,39] can be represented as their difference. Enhancing bright objects of interest on a dark background, like bright blood vessels acquired from an image complement of a luminance channel image where the background turns dark, is done by representing the interaction between the image and a structuring element of determined size and form using Top-Hat morphological operation. To detect the small intensity fluctuations, we are using optimized Top-Hat transform and homomorphic filtering on $OAGF^{Y}_{(t,d_x,\epsilon)}$ as described in Equation (10).

$$ToH^{Y} = O\hat{A}GF - \left(O\hat{A}GF \circ ST_o\right) \cdot ST_c$$ (10)

where, $ToH^{Y}$ indicates optimized Top-Hat transform, $ST_o$ and $ST_c$ are structural elements of opening $'\circ'$ and closing $'\cdot'$ respectively.

To further enhance vascular structures, Homomorphic Filtering ($HMF$) [40,41] is applied. It operates in the logarithmic domain to suppress low-frequency illumination components while amplifying high-frequency details of retinal image as given in Equation (11).

$$ToH^{Y} = I^{ill}_{(x,y)} \cdot I^{ref}_{(x,y)}$$ (11)

where, $I^{ref}_{(x,y)}$ represents the reflectance component, containing fine details like blood vessels and $I^{ill}_{(x,y)}$ is obtained from Equation (7).

We further process Equation (11) with Fourier transform as given in Equation (12).

$$F(u,v) = \mathcal{F}\left\{ln(I^{ill}_{(x,y)})\right\} + \mathcal{F}\left\{ln(I^{ref}_{(x,y)})\right\}$$

(12)

where, $\mathcal{F}$ represents the Fourier transform and $F(u,v)$ is frequency representation of the image.

The response of homomorphic filter is given in Equation (13).

$$HMF_{(u,v)} = (\Upsilon_H - \Upsilon_L)\left[1 - e^{-\frac{\vartheta^2(u,v)}{2\rho^2}}\right] + \Upsilon_L$$

(13)

where, $\vartheta(u,v)$ is the distance from the frequency origin, $\rho$ controls the cut-off frequency of the filter, $\Upsilon_H$ and $\Upsilon_L$ are gain parameters. $\Upsilon_H > 1$ enhances high-frequency details and $\Upsilon_L < 1$ reduces low-frequency illumination effects.

Homomorphic filtering in the frequency domain enhances the visibility of vessels by dissociating the reflectance and illumination parts of the image. Vessel structures normally have high-frequency attributes, while background illumination and non-uniform lighting have low-frequency elements. By using a high-pass filtering in the logarithmic domain (as indicated by Equation (13)), the technique inhibits low-frequency effects of lighting and enhances high-frequency reflectance information, including vascular structure edges. All this gives rise to a greater vessel contrast, facilitating more precise segmentation. The selective filtering in the frequency domain is responsible for sustaining the continuity of fine vessels even in poor lighting conditions.

With the help of Equations (12) and (13), the filter response in frequency domain is given in Equation (14).

$$H\hat{M}F_{(u,v)} = HMF_{(u,v)}.F(u,v)$$

(14)

The desired segmented retinal image is obtained by doing inverse log transform followed by inverse Fourier transform with mask as given in Equation (15).

$$I^{segmentation}_{(x,y)} = e^{F^{-1}\left\{H\hat{M}F_{(u,v)}\right\}}.M_{(x,y)}$$

(15)

where, $M_{(x,y)}$ represents the mask used to extract the desired retinal image as shown in Fig 1.

### 3.3. Stage 3: Enhanced image reconstruction

The final stage focuses on reconstructing and enhancing the processed image for diagnostic use. The reconstructed image combines the segmented features with chrominance components to produce a visually enhanced output. As shown in Fig 1, YCbCr to RGB conversion followed by the mask operation produces an enhanced image given in Equation (16).

$$I^{Enhanced}_{(x,y)} = YCbCr(OAGF^Y_{(t,d_x,\epsilon)},\ I^{Cb}_{(x,y)}, I^{Cr}_{(x,y)}).M_{(x,y)}$$

(16)

This framework provides a robust solution for processing retinal fundus images by integrating illumination correction, segmentation, and enhancement techniques. The first stage ensures even illumination and contrast enhancement using the OAGF filter. The second stage extracts key diagnostic features like blood vessels with high precision using the optimized Top-Hat transform and homomorphic filtering. The final stage reconstructs the image into a visually enhanced format, combining luminance and chrominance components for diagnostic readiness.

## 4. Experimental results discussion

### 4.1. Experiment setup

To evaluate the performance of the proposed Optimal Anisotropic Guided Filtering (OAGF) framework, we have conducted experiments on publicly available retinal fundus imaging datasets, including the DRIVE [42] and STARE [43] datasets. The DRIVE dataset comprises of 40 high-resolution fundus images with corresponding ground truth annotations for vessel segmentation, while the STARE dataset includes 20 images with expert-annotated masks.

The framework was implemented in MATLAB R2024a, and experiments were performed on a workstation equipped with an NVIDIA RTX 3090 GPU, 32 GB RAM, and an Intel Core i9-12900K CPU. Both peak and average memory consumption while processing do not exceed 2 GB, rendering the approach feasible for use in virtually all contemporary GPU-powered machines found in clinical imaging settings. GPU parallelization works greatly in its favor in those parts of OAGF that include convolution (homomorphic filtering in the frequency domain) and structural patch calculations. We used MATLAB's GPU support for accelerations in Fourier transforms and matrix multiplications explicitly. Its consistency in terms of performance for various resolutions of typical fundus image data (565×584 for DRIVE, 700×605 for STARE) validates its scalability.

A comparison of the proposed OAGF-based segmentation with four contemporary approaches is conducted. These are Morphological and Double Thresholds Filtering (MDTF) [44], Inception-Like U-Net (IUnet) [45], Retinal Blood Vessels Tortuosity Hybrid Segmentation (RVTHS) [46] and Multiscale Feature Fusion based Segmentation (MFFS) [47]. Similarly, for OAGF based enhancement, five different approaches are used. These are Adaptive Pixel Clustering (APC) [48], Contrast-Limited Adaptive (CLA) Histogram Equalization [49], Non-Subsampled Contourlet Transform (NSCT) [50], Discrete Wavelet Transform (DWT) [51] and Curvelet Transform (CT) [52].

The performance evaluation of the suggested OAGF technique for segmentation and enhancement of the retinal images has been assessed both quantitatively and qualitatively. For these existing methods, we have followed the author's recommendation for the default parameters. We used the standard binary segmentation formulas for the commonly used metrics in retinal image segmentation: Pixel Accuracy (PA), Mean Accuracy (MA), Intersection over Union (IoU), Dice Coefficient (DC), Precision (PR), Recall (RE), and F1 Score (F1S) [53,54]. Similarly, we used six quality measures for quantitative analysis of retinal fundus enhancement. These are Blind/Reference-less Image Spatial Quality Evaluator (BRISQUE) [55], Naturalness Image Quality Evaluator (NIQE) [56], Perception based Image Quality Evaluator (PIQUE) [57], Iterative Minimum Mean Square Error (IMMSE) [58], Peak-Signal-to-Noise Ratio (PSNR) and Structural Similarity Index Measure (SSIM) [59].

### 4.2. Quantitative analysis

The following are the terms used in quantitative analysis explained in this section.

*TP = True Positives* (predicted vessel pixel that is actually vessel)
*FP = False Positives* (predicted vessel pixel that is actually background)
*TN = True Negatives* (predicted background pixel that is actually background)
*FN = False Negatives* (predicted background pixel that is actually vessel)

**4.2.1. Pixel Accuracy (PA).** Pixel Accuracy is the ratio of the number of correctly classified pixels (both vessel and background) to the total number of pixels in the image as given in Equation (17).

$$PA = \frac{TP + TN}{TP + FP + TN + FN}$$

(17)

**4.2.2 Mean Accuracy (MA).** Mean Accuracy is the average of the per-class accuracies. In a binary segmentation, it is the mean of the accuracy on the vessel class and the accuracy on the background class as given in Equation (18).

$$MA = \frac{1}{2}\left(\frac{TP}{TP+FN} + \frac{TN}{TN+FN}\right)$$

(18)

where, $\frac{TP}{TP+FN}$ is the accuracy on the vessel class and $\frac{TN}{TN+FN}$ is the accuracy on the background class.

**4.2.3. Intersection over Union (IoU).** Intersection over Union (IoU), also known as the Jaccard index, measures the overlap between the predicted segmentation and the ground truth relative to their union as given in Equation (19).

$$IoU = \frac{TP}{TP+FN+FP}$$

(19)

**4.2.4. Dice Coefficient (DC).** The Dice Coefficient, or Sørensen–Dice index, quantifies the similarity between the predicted segmentation and the ground truth by measuring the overlap between them as shown in Equation (20).

$$DC = \frac{2 \times IoU}{1 + IoU}$$

(20)

DC is numerically equal to the F1S for a single foreground class if 'Precision' and 'Recall' are calculated on the same set of predictions.

**4.2.5. Precision (PR).** Precision (also known as Positive Predictive Value) is the proportion of pixels predicted as vessel that are truly vessel pixels given in Equation (21).

$$PR = \frac{TP}{TP+FP}$$

(21)

**4.2.6 Recall (RE).** Recall (also known as Sensitivity) is the proportion of true vessel pixels that were correctly identified by the segmentation as given in Equation (22).

$$RE = \frac{TP}{TP+FN}$$

(22)

**4.2.7. F1 Score (F1S).** The F1 Score is the harmonic mean of 'Precision' and 'Recall'. It provides a single measure that balances both the false positives and false negatives as given in Equation (23).

$$F1S = 2 \times \frac{PR \times RE}{PR + RE}$$

(23)

**4.2.8. Blind/Reference-less Image Spatial Quality Evaluator (BRISQUE).** This metric is a widely used image quality assessment metric with no reference image that computes quality scores without requiring a reference image. It relies on locally normalized luminance coefficients from the Natural Scene Statistics (NSS). The BRISQUE combines multiple mathematical steps into a framework, but it can be summarized conceptually as given in Equation (24).

$$BRISQUE = SVR\left(f\left(\hat{I}, GGD, AGGD\right)\right)$$

(24)

where, $\hat{I} = \frac{I(i,j) - \mu(i,j)}{\sigma(i,j) + C}$ is a locally normalized luminance coefficient, $\mu(i,j)$ and $\sigma(i,j)$ are local mean and standard deviation, respectively. *GGD* is Generalized Gaussian Distribution used to model normalized coefficients. *AGGD* is Asymmetric

Generalized Gaussian Distribution used to model pairwise neighboring coefficients. $f$ is a function of feature vector extracted from $GGD$ and $AGGD$ parameters across luminance and pairwise orientations. $SVR(\cdot)$ is Support Vector Regression model that maps the feature vector $f$ to a predicted image quality score.

**4.2.9. Naturalness Image Quality Evaluator (NIQE).** Using a straightforward and effective space domain Natural Scene Statistic (NSS) methodology, this measure is based on a "quality aware" set of statistical features. These features are extracted from a collection of original, undistorted images. This can be summarized mathematically as given in Equation (25).

$$NIQE = d\left(f_{test},\ f_{model}\right) = \sqrt{\left(f_{test} - f_{model}\right)^T \Sigma^{-1}\left(f_{test} -\ f_{model}\right)}$$
(25)

where, $f_{test}$ is feature vector of the test image derived from NSS, $f_{model}$ is mean feature vector of a pristine natural image model, $d(\cdot,\ \cdot)$ is Mahalanobis distance, $\Sigma$ is covariance matrix of the NSS features in the pristine natural image model.

**4.2.10. Perception based Image Quality Evaluator (PIQE).** This metric represents the average perceptual distortion across the image, with lower scores indicating better image quality. Blocks with uniform regions are ignored to focus on areas with perceptual significance as given in Equation (26).

$$PIQUE = \frac{\left(\sum_{j=1}^{N_{SA}} D_{sj}\right) + K_1}{\left(N_{SA} + K_1\right)}$$
(26)

where, $N_{SA}$ represents spatially active blocks in a given image. To avoid numerical instability when the denominator approaches zero, a positive constant $K_1$ is incorporated. $D_{sj}$ is a variance feature that is used to determine the degree of distortion in a distorted block.

**4.2.11. Iterative Minimum Mean Square Error (IMMSE).** This metric follows an iterative algorithm designed to estimate an image from noisy or degraded observations. IMMSE minimizes the Mean Square Error (MSE) while balancing fidelity to the observed data and consistency with prior estimates as given in Equation (27).

$$\hat{I}_{(k+1)} = \arg\min_I \left( \|Y - H \cdot I\|_F^2 + \lambda \left\| I - \hat{I}_{(k)} \right\|_F^2 \right)$$
(27)

where, $\hat{I}_{(k+1)}$ is the updated image estimate at the $(k+1)^{th}$ iteration, $\hat{I}_{(k)}$ is the current image estimate at the $k^{th}$ iteration, $\|Y - H \cdot I\|_F^2$ is data fidelity term representing the Frobenius norm of the difference between the observed image $Y$ and its degraded version $H \cdot I$. This enforces alignment with the observed image data. $\left\| I - \hat{I}_{(k)} \right\|_F^2$ is a regularization term penalizing large deviations of the current estimate $I$ from the prior estimate $\hat{I}_{(k)}$ ensuring smoothness or consistency and $\lambda$ is regularization parameter that regulates the balance between data fidelity and prior consistency.

**4.2.12. Peak-Signal-to-Noise Ratio (PSNR).** This metric is frequently employed in the field of image and video processing to evaluate the quality of compressed, reconstructed, or denoised images in comparison to their original versions. By comparing the similarity between processed image and its original version, it quantifies the degree of distortion or degradation that is introduced during processing. The image quality is quantified in decibels (dB), offering a logarithmic assessment as indicated in Equation (28).

$$PSNR = 10\ log_{10}\left(\frac{max^2}{MSE}\right)$$
(28)

where, $max$ indicates maximum intensity value of an image pixel (example: for a 8-bit image $max$ = 255). $MSE = \frac{1}{M\ X\ N} \sum_{i=1}^{M} \sum_{j=1}^{N} \left(I(i,j) - \hat{I}(i,j)\right)^2$, $I(i,j)$ is original image with $M\ X\ N$ as its size, $\hat{I}(i,j)$ is reconstructed image.

**4.2.13. Structural Similarity Index Measure (SSIM).** This metric assesses the similarity between two images, typically for evaluating image quality or degradation. It compares contrast, luminance, and structure between two images. The SSIM is defined mathematically as given in Equation (29).

$$SSIM_{(x,y)} = \frac{(2\mu_x\mu_y + C_1)(2\sigma_{xy} + C_2)}{(\mu_x^2 + \mu_y^2 + C_1)(\sigma_x^2 + \sigma_y^2 + C_2)}$$

(29)

where, $x$ and $y$ are two image patches or corresponding pixels from the two images being compared, $\mu_x$ and $\mu_y$ are mean intensities of $x$ and $y$ respectively, $\sigma_x^2$ and $\sigma_y^2$ are variances of $x$ and $y$ respectively, $\sigma_{xy}$ is covariance between $x$ and $y$, $C_1$ is a constant to stabilize the division when $\mu_x^2 + \mu_y^2$ is close to zero and $C_2$ is a constant to stabilize the division when $\sigma_x^2 + \sigma_y^2$ is close to zero.

## 4.3. Results

**4.3.1. Segmentation results analysis.** This section provides an analysis of results derived from existing methodologies in comparison with the proposed OAGF method. From Fig 2, the first and third rows show six retinal fundus images labelled TS-1 to TS-6 and these images represent a variety of retinal images with differences in contrast, illumination, and clarity. The variations in the images are intended to evaluate the robustness of the segmentation algorithm under diverse situations. TS-1 to TS-3 are clearer images with varying levels of vessel visibility and TS-4 to TS-6 are dimmer or more challenging images with uneven illumination and reduced clarity.

The rows Segmentation-1 to Segmentation-6 present the segmentation results corresponding to each test case. Each output focuses on extracting the vascular structure of the retina. Segmentation-1 clearly extracts the main vascular tree of the retina, including both major and minor vessels. Most importantly, vessel bifurcations are well-preserved, and there are minimal false edges.

Similar to output labelled as Segmentation-1, Segmentation-2 output captures the vascular structure with good accuracy and vessel connectivity remains consistent. The proposed OAGF performs well even with slight illumination variations in TS-2, successfully segmenting smaller vessels. The proposed OAGF algorithm performs exceptionally well on high-contrast, well-illuminated images (TS-1 and TS-2) and it consistently captures the vascular tree with high accuracy, including finer branches. Segmentation-3 and segmentation-4 captures large vessels effectively but struggles with finer details. Noise and artifacts are present, likely due to the uneven illumination and reduced image quality in TS-3 and TS-4. The challenges observed indicate the algorithm's sensitivity to image quality, suggesting potential improvement areas in pre-processing or vessel enhancement.

Large vessels are partially segmented but lack the clarity observed in Segmentation-1 and Segmentation-2. In segmentation-5 and segmentation-6, the vascular tree is reasonably well-segmented, capturing both large and some smaller vessels and compared to Segmentation-4, the result is cleaner. The performance here suggests that the algorithm handles medium-quality images like TS-5 better than dimmer cases like TS-4 or TS-6. For medium-quality inputs (TS-5), the algorithm still delivers acceptable segmentation, retaining major vessel structures and reducing noise.

Fig 3 presents a comparative analysis of different segmentation techniques applied to a retinal fundus image, aiming to extract blood vessels with high accuracy. The original fundus image Fig 3(i) serves as the reference, containing detailed blood vessel structures along with background retinal features. Fig 3(ii) applies morphological filtering to enhance vessels but results in a sparse, fragmented extraction, losing finer vessel details. Fig 3(iii) a deep learning-based segmentation approach, performs better than MDTF by capturing more vessel structures; however, some minor vessels remain indistinct.

Fig 3(iv) identifies a more detailed vascular network but suffers from high background noise, making vessel differentiation challenging. Fig 3(v) MFFS integrates multiple feature extraction techniques but produces a faint segmentation with many missing vessels, leading to an incomplete vascular structure. In contrast, in Fig 3(vi) the proposed OAGF outperforms all existing methods. It provides a well-defined, continuous, and noise-free segmentation of retinal blood vessels.

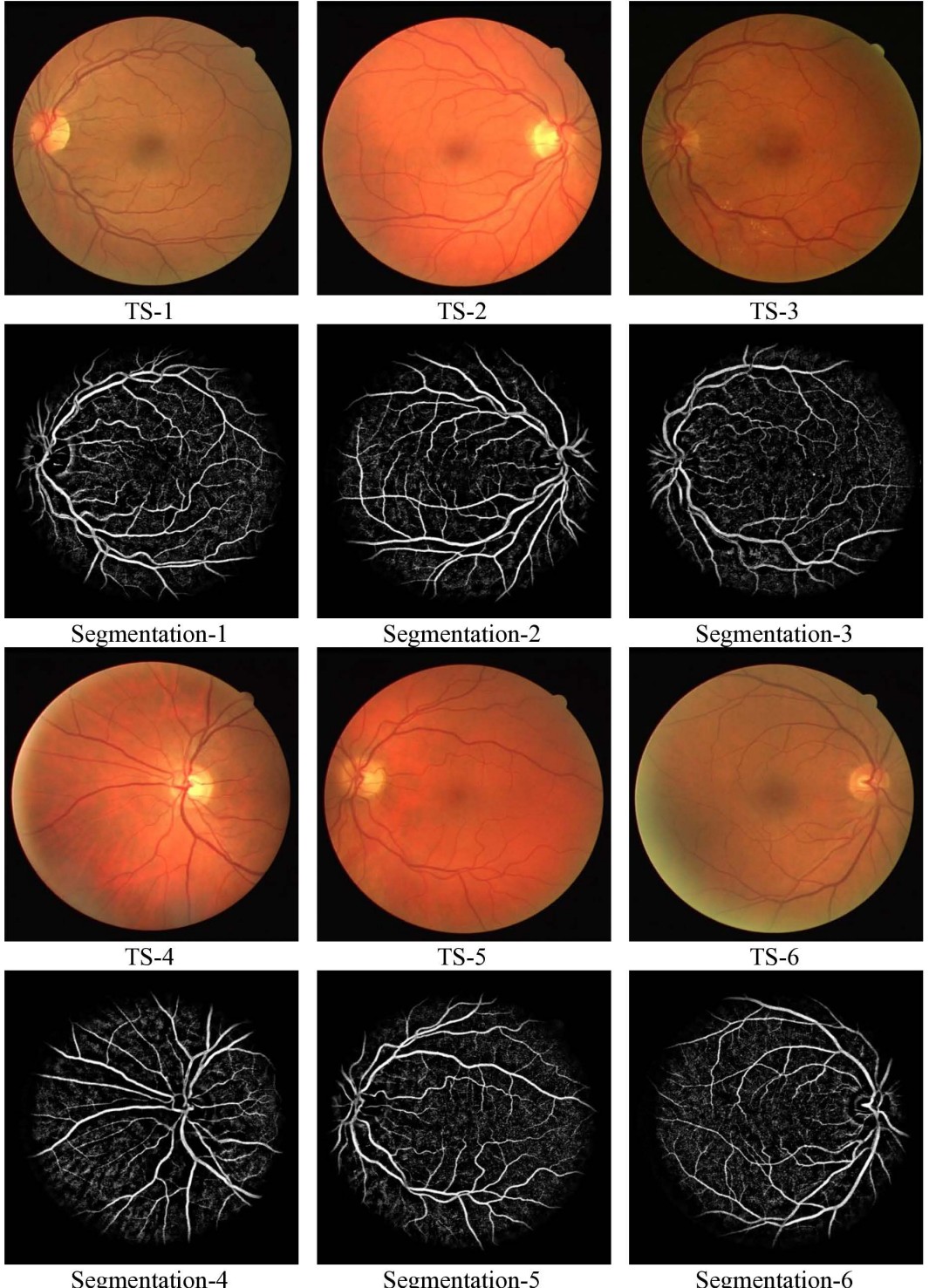

**Fig 2. Retinal image segmentation across six test cases using proposed OAGF.**

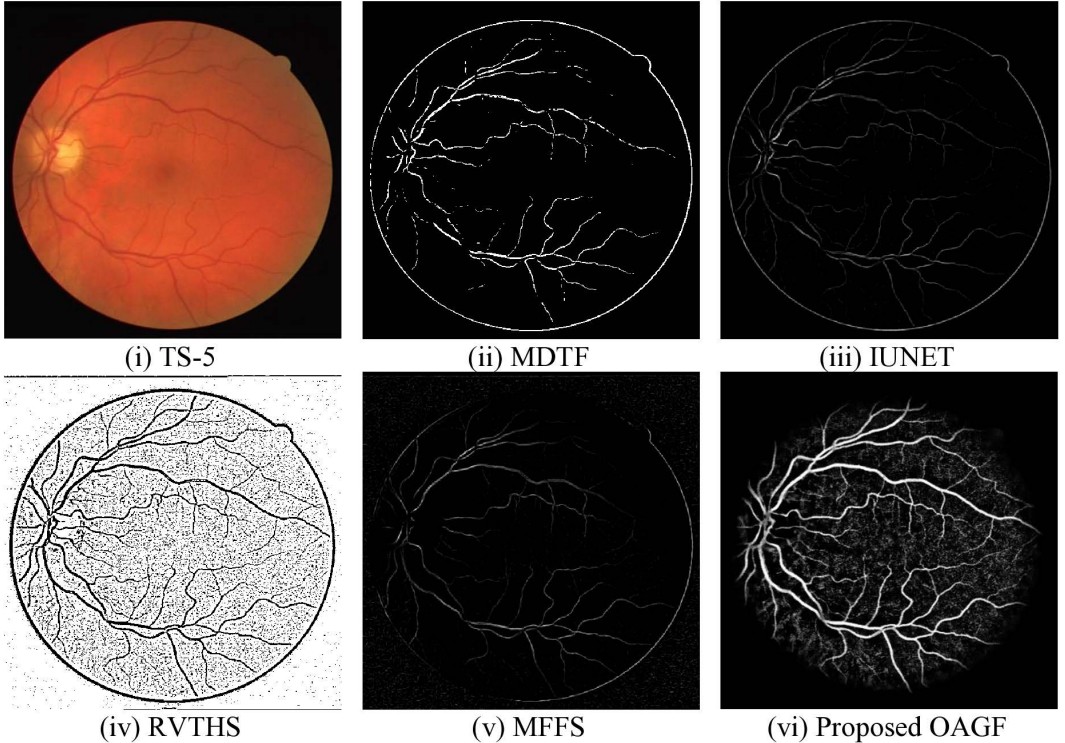

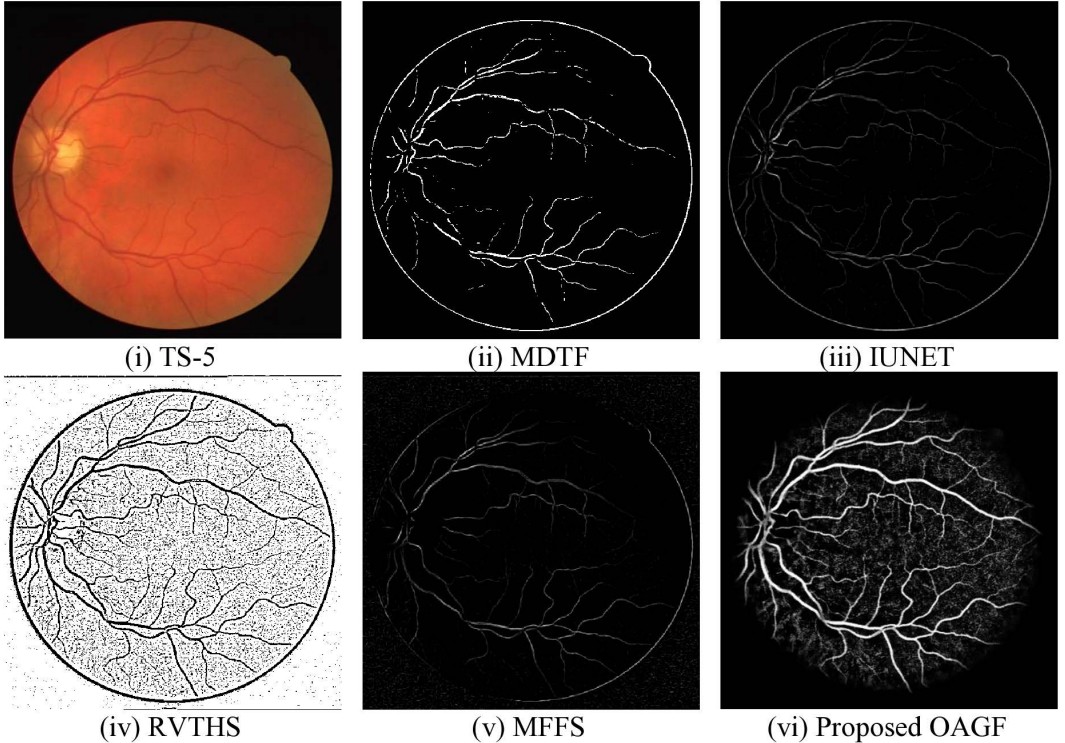

**Fig 3. Subjective assessment of existing and proposed OAGF based segmentation approaches.**

The vessels appear clear, with both primary and fine structures accurately extracted. Unlike other methods, OAGF effectively suppresses background noise while maintaining vessel connectivity, making it the most reliable approach for automated retinal vessel segmentation. These results are crucial for medical applications, such as early diagnosis of diabetic retinopathy and other vascular diseases.

To evaluate the effectiveness of the proposed Optimized Adaptive Gaussian Filtering (OAGF) method for retinal vessel segmentation, we performed a quantitative analysis on two publicly available benchmark datasets: DRIVE and STARE. The quantitative results for both datasets are summarized in Table 4.

**Statistical significance tests:** We have performed paired t-tests and Wilcoxon signed-rank tests on a random subsample of test images of the DRIVE and STARE datasets. We used these tests on segmentation performance metrics: F1 Score (F1S), Dice Coefficient (DC), and Intersection over Union (IoU) in comparison to four baseline methods: MDTF, IUNet, RVTHS, and MFFS, against the proposed method, which is OAGF. Wilcoxon signed-rank test invariably provided **p=0.00195**, which signifies persistent directional superiority of OAGF in all tested case studies. As provided in Table 5, p-values of t-tests were all less than 0.0001. These results verify performance improvement brought about by OAGF not being caused by random variability and being statistically significant.

**Performance on the DRIVE Dataset is as follows:** The proposed OAGF method consistently outperforms existing methods across all evaluation metrics. OAGF achieves the highest PA of 0.971, surpassing all other approaches, indicating its superior ability to correctly classify vessel and non-vessel pixels. Moreover, the MA score of 0.872 confirms that OAGF maintains a high level of accuracy across different classes. The IoU score of 0.755 and Dice Coefficient (DC) of 0.86 demonstrate that the segmented vessels exhibit strong alignment with the ground-truth annotations. Additionally, OAGF achieves the highest precision (0.845) and recall (0.811), suggesting that it effectively captures both large and fine vessel

**Table 4. Quantitative analysis of proposed and existing methods for DRIVE and STARE datasets.**

| | DRIVE dataset | | | | | STARE dataset | | | | |
|--------|-------|-------|-------|------|---------------|-------|-------|-------|-------|---------------|
| Metric | MDTF | IUNet | RVTHS | MFFS | Proposed OAGF | MDTF | IUNet | RVTHS | MFFS | Proposed OAGF |
| PA | 0.952 | 0.965 | 0.948 | 0.963 | **0.971** | 0.952 | 0.942 | 0.931 | 0.958 | **0.962** |
| MA | 0.832 | 0.825 | 0.839 | 0.847 | **0.872** | 0.812 | 0.841 | 0.82 | 0.854 | **0.868** |
| IoU | 0.689 | 0.744 | 0.701 | 0.741 | **0.755** | 0.676 | 0.737 | 0.698 | 0.723 | **0.745** |
| DC | 0.816 | 0.853 | 0.824 | 0.851 | **0.86** | 0.807 | 0.848 | 0.822 | 0.839 | **0.854** |
| PR | 0.712 | 0.781 | 0.795 | 0.791 | **0.845** | 0.796 | 0.787 | 0.738 | 0.791 | **0.834** |
| RE | 0.739 | 0.784 | 0.725 | 0.769 | **0.811** | 0.749 | 0.775 | 0.796 | 0.784 | **0.801** |
| F1S | 0.725 | 0.782 | 0.758 | 0.779 | **0.827** | 0.772 | 0.781 | 0.765 | 0.787 | **0.817** |

PA: Pixel Accuracy, MA: Mean Accuracy, IoU: Intersection over Union, DC: Dice Coefficient, PR: Precision, RE: Recall, F1S: F1 score.

**Table 5. Statistical significance test analysis.**

| | t-test p | | |
|----------------|----------|---------|---------|
| Methods | F1S | DC | IoU |
| OAGF vs MDTF | 0.00001 | 0.00002 | 0.00002 |
| OAGF vs IUNet | 0.00002 | 0.00009 | 0.00009 |
| OAGF vs RVTHS | 0.00001 | 0.00001 | 0.00001 |
| OAGF vs MFFS | 0.00001 | 0.00001 | 0.00001 |

structures while minimizing false positives and false negatives. The F1 Score (0.827) further validates that OAGF maintains an optimal balance between precision and recall, reinforcing its robustness in vessel segmentation.

***Performance on the STARE dataset is as follows:*** The OAGF method again achieves the highest accuracy across all metrics, demonstrating its generalizability and effectiveness in segmenting retinal vessels from different sources. Specifically, OAGF attains a PA of 0.962 and a MA of 0.868, outperforming other techniques. The IoU (0.745) and DC (0.854) scores reinforce the robustness of OAGF, ensuring that the segmented vessels have a high degree of overlap with the ground-truth vessel structures. Furthermore, Precision (0.834) and Recall (0.801) remain the highest, confirming that OAGF effectively captures vessel structures while minimizing segmentation errors. The F1 Score (0.817) further highlights the superiority of OAGF in maintaining a strong balance between sensitivity and specificity.

As with the DRIVE dataset, IUNet achieves the second-best results but remains inferior to OAGF, particularly in IoU and DC scores. MFFS and RVTHS display moderate performance, while MDTF continues to struggle with segmentation accuracy particularly in preserving vessel continuity.

The quantitative analysis demonstrates that the proposed OAGF method outperforms all existing vessel segmentation techniques across both DRIVE and STARE datasets. OAGF effectively balances precision and recall while achieving the highest accuracy in terms of IoU and Dice Coefficient, ensuring superior vessel segmentation performance. These findings suggest that OAGF is a promising approach for accurate and reliable automated retinal vessel segmentation, making it highly suitable for medical applications such as ophthalmic disease diagnosis and retinal image analysis.

Further analysis made by graphical representation with standard deviation, as shown in Fig 4 indicate that the proposed OAGF method consistently outperforms all other approaches in both datasets, achieving the highest PA and MA scores, indicating its superior ability to accurately classify vessel and non-vessel pixels. Additionally, the IoU and DC scores of OAGF are the highest, demonstrating a strong overlap between the segmented vessel structures and ground truth annotations. The method also excels in Precision and Recall, ensuring that it effectively detects vessel structures while minimizing false positives and false negatives.

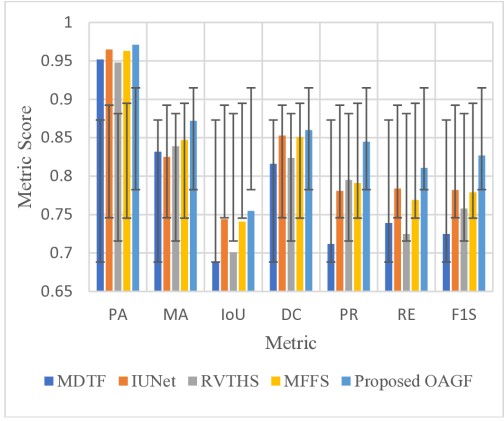
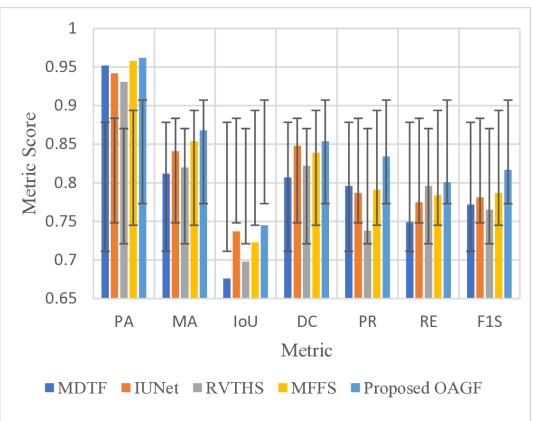

| (i) Metrics evaluation with DRIVE dataset | (ii) Metrics evaluation with STARE dataset |

**Fig 4. Metrics Evaluation with DRIVE and STARE Datasets.**

The highest F1 Score (F1S) further confirms that OAGF maintains an optimal balance between precision and recall, leading to superior segmentation results. While IUNet performs relatively well and is the second-best method, it still lags behind OAGF in vessel continuity and detail preservation. In contrast, traditional methods such as MDTF and RVTHS exhibit weaker performance, particularly in recall and IoU, indicating their limitations in capturing fine vascular structures. These results demonstrate that OAGF is a more reliable and generalizable segmentation approach, making it highly suitable for automated retinal vessel segmentation in medical imaging applications.

**4.3.2. *Failure case analysis*.** Fig 5 depicts a typical case of failure when the presented method of OAGF performs poorly in the low-contrast region of the retina. Thin vessels are obvious in the ground truth within the zoomed region (the red box) but are missing or fragmented in the output of OAGF. The probable reasons are low light, vessel-background ambiguity, and lack of enough gradient magnitude to initiate directional diffusion. Although OAGF is able to enhance dominant vessels significantly, this case reflects one of the shortcomings of the present method in detecting very faint micro-vessel structures. Potential future improvements involve the implementation of adaptive vessel confidence models or learning-based post-processing components.

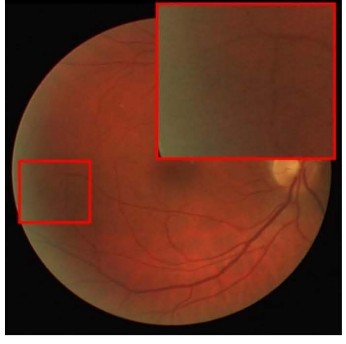
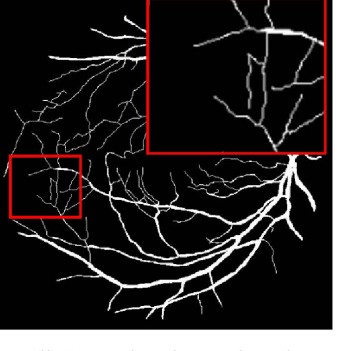
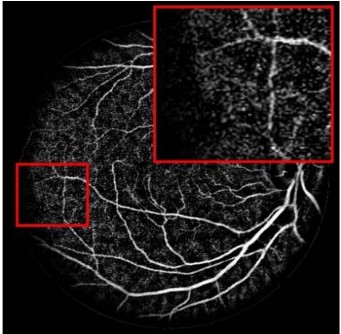

|     (i) Source fundus image     |     (ii) Ground truth vessel mask     |     (iii) Proposed OAGF segmentation     |

**Fig 5. Visual failure case comparison.** Red box shows zoomed region where thin vessels are either missed or disconnected in the proposed method due to low contrast and anatomical ambiguity.

**4.3.3. *Ablation study and component-level analysis*.** To quantify the individual contributions of each component in the proposed framework, we conducted an ablation study on the DRIVE and STARE datasets. The configurations tested include the full pipeline ($OAGF + ToH^Y + HMF$), and variants where each module was removed independently. As indicated in Table 6, the complete model performed best on all measures with a DC of 0.860 on DRIVE dataset and 0.854 on STARE dataset, validating the interaction of the three elements.

Removal of OAGF resulted in the maximum drop in performance, with F1 Scores declining to 0.781 (DRIVE) and 0.772 (STARE), indicating OAGF's essential function for structural adaptivity and noise attenuation. Deletion of optimized Top-Hat transform resulted in a moderate drop, especially in IoU, due to weakened enhancement of finer vascular details. Omission of Homomorphic Filtering resulted in minor drops in both datasets, reflecting its importance in normalizing global illumination and contrast, especially under poor light conditions.

This verifies that each of these modules contributes significantly towards the ultimate outcome, and their fusion results in strong segmentation and enhancement under varying conditions of the retinal image.

**4.3.4. *Hyperparameters (t, dx, ε) sensitivity analysis*.** Fig 6 shows the analysis of how the performance of the employed OAGF method is influenced by three important hyperparameters: the directional regularization parameter ($dx$), the stopping time of iterations ($t$), and the smoothness regularization parameter ($\epsilon$). The analysis is carried out using three widely used segmentation metrics: F1S, DC, and IoU. The sensitivity study is important to comprehend the stability and the need to tune OAGF under different imaging conditions.

Fig 6(i) shows how changing dx, which determines directional regularization during filtering, affects the performance. As can be seen, performance in segmentation increases as dx rises between 0.001 and 0.01, where the three metrics are optimal at $dx$ =0.01. Beyond this, performance deteriorates, especially at $dx$ =0.05 and 0.1, due to over-directional smoothing causing blurring of vessel boundaries and loss of the finer details. Extremely small values of $dx$ perform poorly as well because they do not enforce structural constraints effectively enough.

Fig 6(ii) depicts the sensitivity to the iteration parameter $t$, which controls the stopping time of the process of anisotropic diffusion. The findings are such that $t$ =1 or $t$ =2 provides the best performance, and $t$ greater than 3 gradually decreases segmentation accuracy. The reason is over-smoothing, which destroys small vessel structures and weak edges, which are of paramount importance in the analysis of retinal images.

Fig 6(iii) shows the effect of the regularization parameter $\epsilon$, which determines the degree of smoothness within the optimization framework. Optimal performance occurs at $\epsilon$ = 0.001, which gives the best trade-off between enhancement of vessels and background noise removal. Lower values result in background noise which is not removed sufficiently, whereas higher values result in blurred boundary contours of vessels and performance loss in all measured metrics.

**4.3.5 Enhancement results analysis.** Fig 7 illustrates a comparative evaluation of retinal image enhancement techniques, including existing methods ((ii) to (vi)) and a proposed method (vii). The comparison uses a source image (i) and highlights a specific region (red box) to emphasize local enhancements. In Fig 7(i), the original retinal image has poor brightness and contrast and critical features, such as blood vessels and the optic disk, are not clearly visible. It serves as a baseline for evaluating the enhancement performance of other methods.

**Table 6. Ablation Study and Component-Level Analysis.**

| Configuration | DRIVE dataset | | | STARE dataset | | |
|---|---|---|---|---|---|---|
| | DC | IoU | F1S | DC | IoU | F1S |
| Full Pipeline ($OAGF + ToH^Y + HMF$) | 0.86 | 0.755 | 0.827 | 0.854 | 0.745 | 0.817 |
| Without OAGF | 0.819 | 0.701 | 0.781 | 0.808 | 0.678 | 0.772 |
| Without $ToH^Y$ | 0.833 | 0.72 | 0.798 | 0.821 | 0.696 | 0.787 |
| Without $HMF$ | 0.827 | 0.713 | 0.793 | 0.819 | 0.693 | 0.782 |

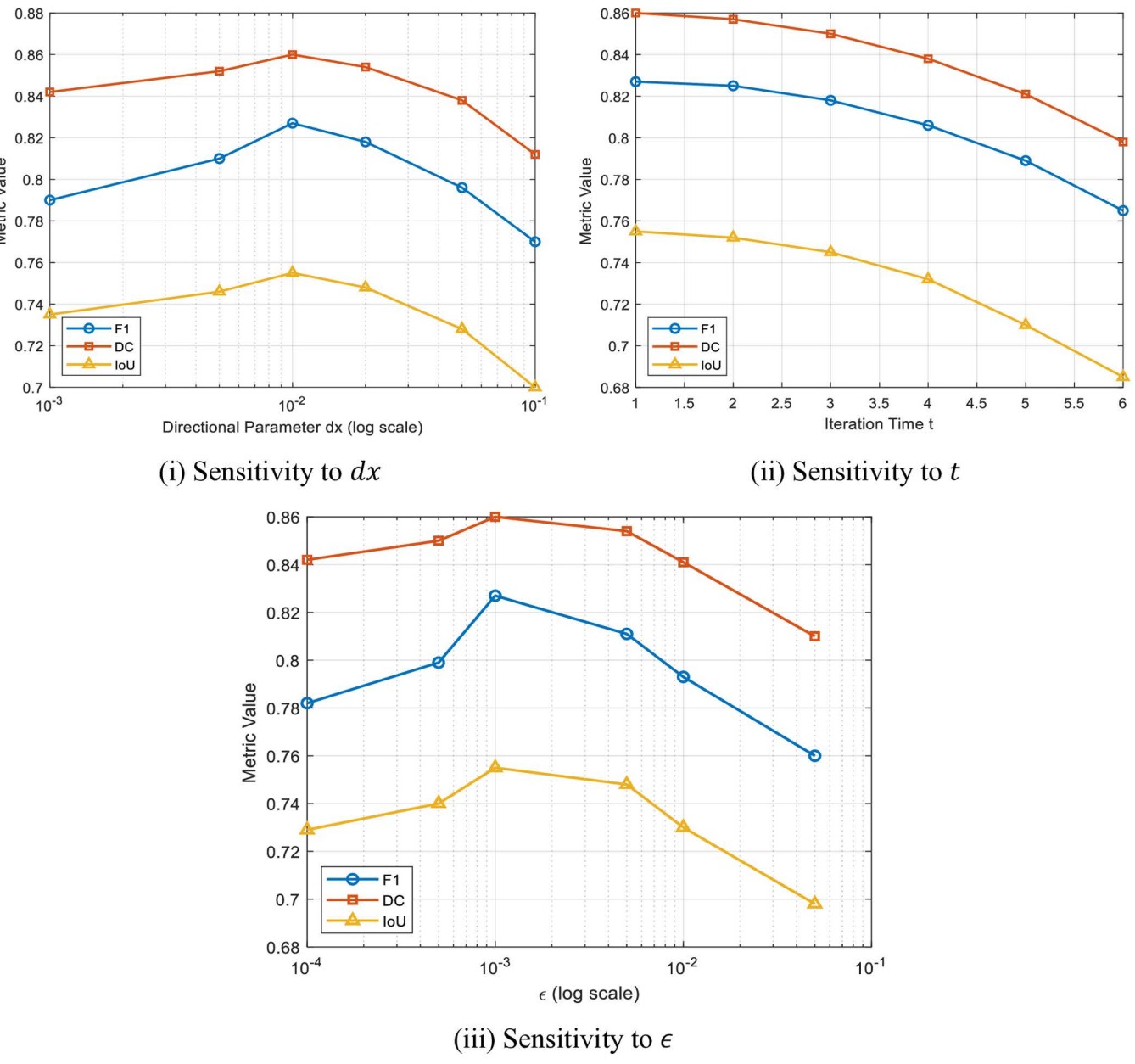

(i) Sensitivity to $dx$

(ii) Sensitivity to $t$

(iii) Sensitivity to $\epsilon$

**Fig 6. Sensitivity analysis of OAGF to key hyperparameters.**

In Fig 7(ii), the APC method enhances overall brightness and contrast, making some features more discernible compared to the source image but it introduces noticeable artifacts and noise, reducing the image's clinical value. Since the enhancement is uneven, the fine details like small blood vessels and subtle textures are not well-preserved.

In Fig 7(iii), the CLA method significantly improves contrast, making retinal structures such as blood vessels more visible and the optic disk is enhanced, aiding in feature recognition, but over enhances certain regions, leading to unnatural brightness and reduced clinical interpretability. Subtle details may be lost due to overemphasis on contrast.

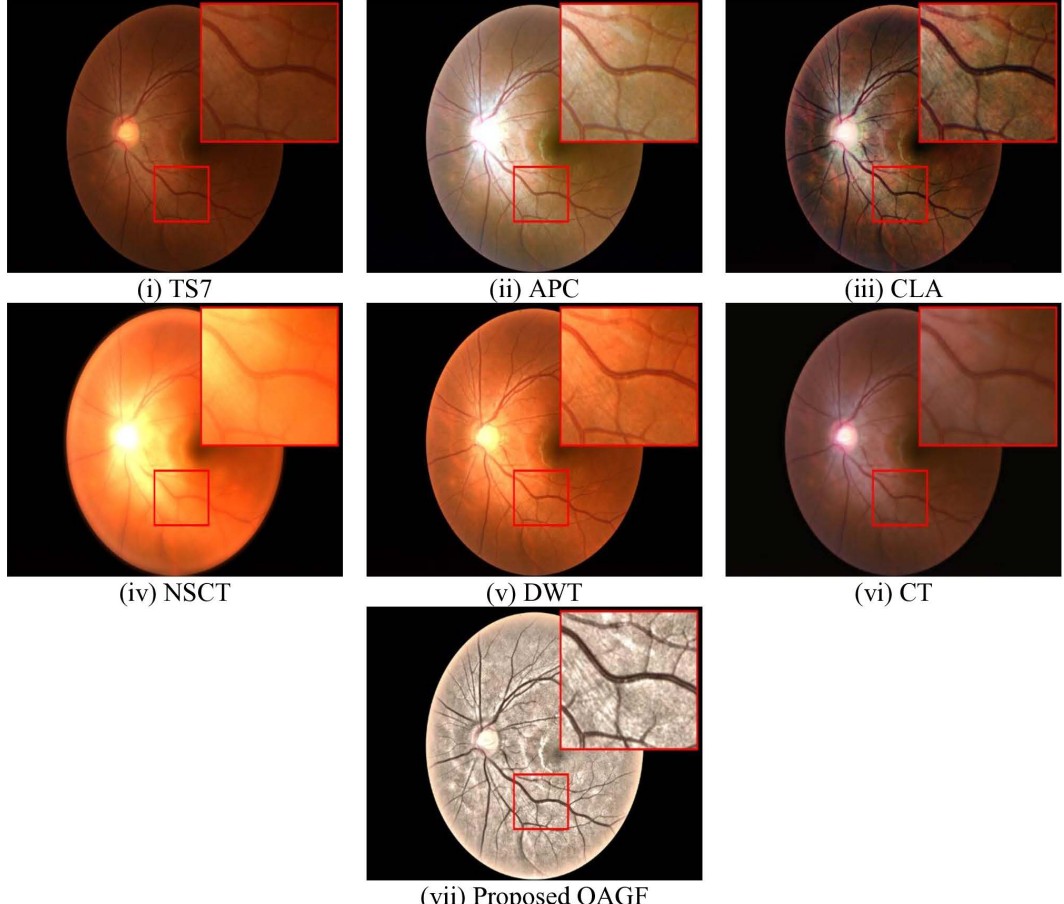

(i) TS7 (ii) APC (iii) CLA

(iv) NSCT (v) DWT (vi) CT

(vii) Proposed OAGF

**Fig 7. Subjective assessment of existing and proposed OAGF based enhancement.**

In Fig 7(iv), the NSCT method focuses on enhancing edges and structural details, particularly around blood vessels. It maintains better balance compared to CLA and APC, resulting in a more natural appearance. Although effective in edge enhancement, it does not significantly improve brightness or contrast globally and some finer details remain less prominent compared to the proposed OAGF method.

In Fig 7(v), the DWT algorithm provides moderate enhancement in contrast and reduces noise, making the image slightly sharper and enhances visibility of medium-sized blood vessels and structural features. The disadvantage of this method is it does not adequately enhance finer structures like capillaries. The overall enhancement is less pronounced compared to NSCT or the proposed OAGF method.

In Fig 7(vi), the CT method excels at enhancing curvilinear structures such as blood vessels, making them more prominent as well as provides good edge enhancement without introducing excessive artifacts. The drawback of this approach is, it is effective for vessel enhancement, but it does not significantly improve the overall brightness or contrast of the retinal image and fails to achieve the fine balance of global and local enhancement seen in the proposed OAGF method.

The proposed OAGF performance assessment is showed in Fig 7(vii). The red-boxed region demonstrates superior detail preservation, highlighting the method's effectiveness in enhancing local features while maintaining global consistency, achieving optimal contrast and brightness enhancement without overexposing or distorting any regions. Blood

vessels, including finer capillaries, are clearly visible with sharp boundaries and provides a natural, balanced appearance, making it ideal for clinical diagnostics.

Unlike APC and CLA, it avoids introducing artifacts or over-enhancing the image. It performs better than NSCT, DWT, and CT by offering both fine detail enhancement and global contrast improvement. It strikes the perfect balance between enhancement and preserving the natural look, making it superior in all aspects. It outperforms all existing methods in terms of visual clarity and detail preservation. This makes the proposed OAGF method a reliable tool for clinical applications, such as detecting retinal diseases, where accurate visualization of fine structures is essential.

The proposed framework of OAGF strikes a good balance between enhancing global contrast and preserving fine detail by utilizing a blend of gradient-aware diffusion and luminance-aware processing. In contrast to the uniform enhancement of the contrast by CLA at the expense of local detail degradation, OAGF performs edge-aware smoothing in the Y channel of the YCbCr color space. This provides selective enhancement of low-intensity vessel regions and avoids excessive enhancement of already bright regions.

Additionally, structural patch-based anisotropic diffusion in OAGF dynamically adjusts the smoothing locally in the direction of gradient directionality, keeping high-contrast boundaries of the vessels and denoising the background regions effectively. Compared to the more conventional approach of fixing the directions in the sub-bands in NSCT, the use of localized, directionally aware filtering by OAGF prevents artifacts of the type of halos and ringing, hence the method is able to foreground-enhance both global and microvascular detail without inducing perceptual distortions.

We used DRIVE and STARE datasets for objective performance analysis with metrics that are widely used to assess image enhancement quality of existing and proposed OAGF methods as listed in Table 7. **Improved performance is indicated by lower values** for BRISQUE, NIQE, PIQE, and IMMSE and **higher values** for PSNR and SSIM. More specifically, BRISQUE values below 30 would be considered good perceptual quality, while values below 20 are regarded as high quality with minimal distortions. NIQE values below 5 appear to depict good naturalness of an image with values below 4 being high visual fidelity. PIQE is a block-based measure of perceptual distortion such that lower scores signify better scores; values below 50 appear to be acceptable with values below 30 indicative of enhanced image clearness. IMMSE is an error-based measure such that values below 0.1 appear to signify highly accurate restorative work with minimal deviation in ideal estimates. PSNR, in general use in image compression as well as enhancement works, is regarded as being acceptable at values above 30 dB with excellent quality at values above 50 dB. Lastly, SSIM values range between 0 and 1 such that scores above 0.95 appear to be regarded as excellent in structural similarity as well as feature conservation

APC and CLA perform inconsistently with moderately high BRISQUE values, suggesting occasional over-enhancement or loss of detail. NSCT, DWT, and CT methods have the highest (worst) BRISQUE scores, especially NSCT, which often results in over-sharpening or noise amplification. Proposed OAGF demonstrates excellent performance, consistently achieving substantially lower BRISQUE scores, such as 1.92 for sample 7 and 6.19 for sample 1, highlighting its robustness in producing high-quality enhancements. The proposed OAGF achieves the lowest BRISQUE scores for all test samples, indicating superior visual quality with reduced artifacts and enhanced naturalness.

APC, CLA, and NSCT methods are generating higher NIQE scores, suggesting the introduction of unnatural textures or poor enhancement quality. DWT and CT methods show slight improvements over NSCT but are still outperformed by OAGF. Similar to BRISQUE, the proposed OAGF outperforms all existing methods with the lowest NIQE values for all test samples, indicating that it produces visually pleasing results close to natural image characteristics.

APC and CLA approaches have moderately high PIQE scores, with CLA performing slightly better than APC in certain cases. NSCT performs the worst, with extremely high PIQE values (e.g., 100.00 for sample 5 and 67.88 for sample 9), indicating poor enhancement with significant perceptual issues. DWT and CT methods have varying results, with DWT performing slightly better but still inferior to OAGF. Proposed OAGF method achieves consistently low PIQE scores, such as 14.27 for sample 5 and 22.63 for sample 10, demonstrating high perceptual quality and effective enhancement.

**Table 7. Comparison of existing and proposed OAGF techniques through objective assessment.**

| Metric | Test Sample | APC | CLA | NSCT | DWT | CT | Proposed OAGF |
|---|---|---|---|---|---|---|---|
| **BRISQUE** | 1 | 16.9517 | 31.0754 | 43.1469 | 42.9758 | 50.4878 | 6.1879 |
| | 2 | 31.2840 | 32.4171 | 48.6040 | 40.4822 | 43.7343 | 19.1113 |
| | 3 | 22.9853 | 30.8478 | 46.2348 | 43.8718 | 43.6839 | 13.7191 |
| | 4 | 20.9187 | 32.3114 | 47.9261 | 43.7280 | 47.6783 | 10.2585 |
| | 5 | 22.1900 | 18.2854 | 47.9727 | 43.6059 | 41.5558 | 9.5337 |
| | 6 | 9.8597 | 6.7156 | 52.3883 | 43.6287 | 47.3240 | 2.9352 |
| | 7 | 8.1298 | 5.8739 | 52.7012 | 43.0576 | 46.8640 | 1.9222 |
| | 8 | 33.8106 | 34.8957 | 47.8206 | 41.6142 | 43.2558 | 21.1063 |
| | 9 | 15.3261 | 31.7096 | 45.7099 | 40.8385 | 32.0534 | 24.4519 |
| | 10 | 22.7766 | 18.0437 | 47.0190 | 40.2923 | 39.4382 | 21.9707 |
| Mean±standard deviation | | 20.42±8.25 | 24.22±11.11 | 47.95±2.88 | 42.41±1.45 | 43.61±5.20 | **13.12±8.20** |
| | **Test Sample** | **APC** | **CLA** | **NSCT** | **DWT** | **CT** | **Proposed OAGF** |
| **NIQE** | 1 | 3.9410 | 5.7222 | 5.2478 | 3.9845 | 4.8610 | 3.6037 |
| | 2 | 3.4851 | 5.3270 | 6.0927 | 3.5735 | 4.4175 | 3.0740 |
| | 3 | 4.1391 | 5.5915 | 5.4812 | 4.0770 | 4.9005 | 3.8606 |
| | 4 | 3.9273 | 5.2223 | 4.9029 | 3.8890 | 4.2525 | 3.7546 |
| | 5 | 4.7605 | 5.0627 | 8.4994 | 4.2164 | 5.0657 | 3.5826 |
| | 6 | 3.8161 | 4.8783 | 4.9875 | 3.7496 | 4.7013 | 3.3508 |
| | 7 | 4.2034 | 5.0233 | 5.5328 | 4.0878 | 5.0656 | 3.6390 |
| | 8 | 4.0655 | 4.8408 | 4.3166 | 3.5108 | 4.0369 | 3.4921 |
| | 9 | 4.2529 | 4.9252 | 6.3421 | 3.8984 | 4.3486 | 3.4224 |
| | 10 | 4.3086 | 5.6045 | 5.8393 | 4.4847 | 4.4363 | 3.5362 |
| Mean±standard deviation | | 4.09±0.34 | 5.22±0.33 | 5.72±1.14 | 3.95±0.29 | 4.61±0.36 | **3.53±0.22** |
| | **Test Sample** | **APC** | **CLA** | **NSCT** | **DWT** | **CT** | **Proposed OAGF** |
| **PIQE** | 1 | 26.3416 | 24.3473 | 16.4488 | 25.2644 | 40.4197 | 33.6771 |
| | 2 | 30.0481 | 28.7369 | 30.8506 | 31.7657 | 38.7272 | 31.0265 |
| | 3 | 29.5947 | 22.0905 | 19.4215 | 25.2537 | 69.1608 | 26.4141 |
| | 4 | 28.0883 | 24.3896 | 24.7422 | 24.9279 | 52.4490 | 26.9930 |
| | 5 | 28.5847 | 24.9634 | 100.0000 | 35.6469 | 54.5770 | 14.2775 |
| | 6 | 27.4983 | 18.7549 | 23.4630 | 18.1018 | 26.6134 | 24.0465 |
| | 7 | 26.8931 | 20.4676 | 22.0696 | 18.3732 | 25.9644 | 22.4911 |
| | 8 | 30.5887 | 31.4794 | 25.9754 | 33.9790 | 47.8198 | 24.9459 |
| | 9 | 33.1595 | 24.1222 | 67.8792 | 38.8727 | 49.3186 | 24.7484 |
| | 10 | 32.5752 | 36.1847 | 31.3423 | 34.3821 | 52.0037 | 22.6358 |
| Mean±standard deviation | | 29.34±2.30 | 25.55±5.24 | 36.22±26.66 | 28.66±7.29 | 45.70±13.18 | **25.13±5.22** |
| | **Test Sample** | **APC** | **CLA** | **NSCT** | **DWT** | **CT** | **Proposed OAGF** |
| **IMMSE** | 1 | 0.3271 | 0.2963 | 0.2991 | 0.2971 | 0.3004 | 0.1277 |
| | 2 | 0.7662 | 0.6938 | 0.6996 | 0.6969 | 0.7013 | 0.0894 |
| | 3 | 0.4485 | 0.4073 | 0.4103 | 0.4083 | 0.4120 | 0.1014 |
| | 4 | 0.4006 | 0.3634 | 0.3670 | 0.3650 | 0.3666 | 0.0990 |
| | 5 | 1.0631 | 0.9622 | 0.9714 | 0.9654 | 0.9697 | 0.0510 |
| | 6 | 0.5471 | 0.4968 | 0.5005 | 0.4989 | 0.5021 | 0.0810 |
| | 7 | 0.5229 | 0.4757 | 0.4789 | 0.4776 | 0.4809 | 0.0846 |
| | 8 | 0.7847 | 0.7098 | 0.7156 | 0.7122 | 0.7173 | 0.0871 |
| | 9 | 1.1097 | 1.0032 | 1.0115 | 1.0067 | 1.0134 | 0.0806 |
| | 10 | 0.7675 | 0.6959 | 0.7018 | 0.6986 | 0.7029 | 0.0814 |

*(Continued)*

**Table 7.** (Continued)

| Metric | Test Sample | APC | CLA | NSCT | DWT | CT | Proposed OAGF |
|---|---|---|---|---|---|---|---|
| Mean±standard deviation | | 0.67±0.27 | 0.61±0.24 | 0.62±0.25 | 0.61±0.24 | 0.62±0.25 | **0.09±0.02** |
| | Test Sample | APC | CLA | NSCT | DWT | CT | Proposed OAGF |
| PSNR | 1 | 14.5324 | 14.4919 | 14.5650 | 14.5371 | 14.4887 | 51.8127 |
| | 2 | 10.8365 | 10.7976 | 10.8739 | 10.8345 | 10.8071 | 52.7795 |
| | 3 | 13.1623 | 13.1102 | 13.1919 | 13.1560 | 13.1173 | 51.7386 |
| | 4 | 13.6531 | 13.6061 | 13.6755 | 13.6432 | 13.6237 | 51.9893 |
| | 5 | 9.4143 | 9.3771 | 9.4488 | 9.4188 | 9.3993 | 53.2854 |
| | 6 | 12.2993 | 12.2485 | 12.3290 | 12.2854 | 12.2582 | 52.1242 |
| | 7 | 12.4955 | 12.4363 | 12.5200 | 12.4748 | 12.4450 | 52.1222 |
| | 8 | 10.7327 | 10.6983 | 10.7760 | 10.7401 | 10.7090 | 52.6539 |
| | 9 | 9.2276 | 9.1962 | 9.2729 | 9.2371 | 9.2081 | 53.2506 |
| | 10 | 10.8290 | 10.7847 | 10.8605 | 10.8234 | 10.7970 | 52.8016 |
| Mean±standard deviation | | 11.72±1.79 | 11.68±1.78 | 11.75±1.78 | 11.71±1.78 | 11.69±1.78 | **52.45±0.57** |
| | Test Sample | APC | CLA | NSCT | DWT | CT | Proposed OAGF |
| SSIM | 1 | 0.4732 | 0.4729 | 0.4738 | 0.4734 | 0.4729 | 0.9477 |
| | 2 | 0.4641 | 0.4637 | 0.4644 | 0.4640 | 0.4638 | 0.9669 |
| | 3 | 0.4728 | 0.4718 | 0.4735 | 0.4727 | 0.4718 | 0.9469 |
| | 4 | 0.4743 | 0.4734 | 0.4749 | 0.4741 | 0.4736 | 0.9523 |
| | 5 | 0.5025 | 0.5021 | 0.5030 | 0.5025 | 0.5023 | 0.9731 |
| | 6 | 0.5074 | 0.5066 | 0.5077 | 0.5072 | 0.5066 | 0.9529 |
| | 7 | 0.5075 | 0.5066 | 0.5077 | 0.5072 | 0.5066 | 0.9533 |
| | 8 | 0.4613 | 0.4610 | 0.4616 | 0.4613 | 0.4611 | 0.9644 |
| | 9 | 0.4500 | 0.4498 | 0.4504 | 0.4500 | 0.4499 | 0.9745 |
| | 10 | 0.4443 | 0.4439 | 0.4445 | 0.4442 | 0.4440 | 0.9671 |
| Mean±standard deviation | | 0.48±0.02 | 0.48±0.02 | 0.48±0.02 | 0.48±0.02 | 0.48±0.02 | **0.96±0.01** |

APC and CLA methods have moderate IMMSE values, indicating reasonable performance but not optimal. NSCT is consistently producing the highest IMMSE values, indicating significant deviation from the ideal enhancement. DWT and CT methods perform slightly better than NSCT but still produce higher IMMSE values compared to the proposed method. Proposed OAGF achieves exceptionally low IMMSE values across all test samples, such as 0.051 for sample 5 and 0.080 for sample 9, indicating minimal error and accurate enhancement.

APC, CLA, NSCT, DWT, and CT methods perform similarly, with PSNR values ranging between 9 and 14 dB across all samples, indicating moderate enhancement quality but limited noise suppression. Proposed OAGF consistently achieves PSNR values exceeding 51 dB, such as 51.81 for sample 1 and 53.28 for sample 5, reflecting minimal noise and excellent enhancement quality. The proposed OAGF achieves significantly higher PSNR values for all test samples compared to the existing methods.

APC, CLA, NSCT, DWT, and CT approaches show minimal variation in SSIM values, generally around 0.45–0.50. These values suggest that these methods provide only moderate preservation of structural information, with noticeable distortions or loss of fine details. The OAGF attains SSIM values exceeding 0.94, including 0.9745 for sample 9 and 0.9731 for sample 5, signifying exceptional structural fidelity and precise enhancement. The OAGF provides the highest SSIM values, reflecting its ability to preserve structural details, contrast, and texture, which are critical for high-quality image enhancement.

The proposed Optimized Adaptive Guided Filter (OAGF) demonstrates clear superiority over existing methods (APC, CLA, NSCT, DWT, and CT) across all metrics and test samples. It effectively enhances retinal images by preserving natural appearance, enhancing fine details, and minimizing distortions, making it the most robust and reliable method for retinal image enhancement.

Although our suggested method produced lower metric values for a few test samples, the average performance of the proposed OAGF demonstrates substantially improved results, as listed in Table 8. The proposed Optimized Adaptive Guided Filter (OAGF) exhibits exceptional performance compared to existing methods (APC, CLA, NSCT, DWT, CT) across all quality metrics. It consistently delivers better perceptual quality, lower distortion, and higher structural fidelity, making it a highly effective method for image enhancement. These results highlight OAGF's reliability and efficiency in improving image quality.

Although the suggested OAGF framework proves to be highly robust and accurate over a range of diverse retinal datasets, potential drawbacks exist. In very low-contrast images, with limited intensity gradients, the conductance model using gradients will not be able to detect the boundaries of vessels in a reliable manner, and the accuracy of segmentation will degrade. A statistical or model-based vessel segmentation method such as the Fast Generalized Linear Model (GLM) approach [60] combines segmentation and denoising using a GLM formulation, demonstrating good vessel continuity under noise.

Likewise, in bright or pathologic artifacts, e.g., exudates or cotton-wool spots, with intensity patterns close to vessels, OAGF will produce false positive segmentation. Finally, for overlapping or tortuously co-entangled vessels, the lack of specific modelling of topology may hinder the ability of OAGF in strictly defining separations between vessels. These drawbacks indicate future potential in using hybrid models that combine structural priors or deep learning-refinement phases, especially for pathologically altered or highly complicated fundus images.

The improved fine retinal vasculature visualization facilitated by the presented OAGF framework is extremely valuable in early disease detection and in monitoring of systemic as well as ocular diseases. In diabetic retinopathy, early signs like microaneurysms, dropout of capillaries, and neovascularization happen at the microvascular scale and can be inadvertently overlooked in poorly enhanced images. In glaucoma, as well, assessment of the retinal nerve fiber layer as well as localized thinning of vessels near the optic disc is improved with increased vessel-edge contrast. In hypertensive retinopathy, fine details such as narrowing of arterioles, arteriovenous nicking, as well as hemorrhages, demand high-fidelity structural detail in order to aid in accurate grading. By maintaining vessel continuity as well as highlighting capillary-level structures, OAGF enhances diagnostic conspicuity as well as allows for improved clinical interpretation.

## 5. Conclusion

This article introduced an Optimal Anisotropic Guided Filtering (OAGF) framework, a comprehensive solution designed specifically for retinal fundus imaging. The OAGF framework integrates enhancement and segmentation into a unified process, leveraging advanced techniques to circumvent the shortcomings of conventional methods. The proposed technique

**Table 8. Comparison of existing and proposed OAGF techniques through objective assessment.**

| Method | BRISQUE | NIQE | PIQE | IMMSE | PSNR | SSIM |
|---|---|---|---|---|---|---|
| APC | 20.4232 | 4.0899 | 29.3372 | 0.6737 | 11.7183 | 0.4757 |
| CLA | 24.2176 | 5.2198 | 25.5536 | 0.6104 | 11.6747 | 0.4752 |
| NSCT | 47.9524 | 5.7242 | 36.2193 | 0.6156 | 11.7513 | 0.4761 |
| DWT | 42.4095 | 3.9472 | 28.6567 | 0.6127 | 11.7150 | 0.4757 |
| CT | 43.6075 | 4.6086 | 45.7054 | 0.6167 | 11.6853 | 0.4753 |
| Proposed OAGF | 13.1197 | 3.5316 | 25.1256 | 0.0883 | 52.4558 | 0.9599 |

operates in three well-defined stages: In the first stage, the RGB source image undergoes illumination correction to normalize uneven lighting, a common issue in retinal images. The luminance (Y) component is extracted by converting the corrected image to the YCbCr color space. This component is processed using anisotropic guided filtering, which effectively enhances image quality by smoothing noise in homogeneous areas while preserving edge details.

The second stage focuses on the segmentation of the retinal vascular tree. An optimized top-hat transform is applied to highlight vascular structures, followed by homomorphic filtering to enhance contrast and extract fine details. This stage ensures that the vascular network is accurately segmented, addressing issues such as spatial inconsistencies and ghost artifacts.

In the final stage, the processed luminance component and the original chrominance components are combined, converting the image back to RGB format. This step produces a visually enhanced image that retains the natural appearance of the retina while emphasizing important structural details.

Extensive evaluations of the OAGF framework were conducted using publicly available databases, comparing its performance against cutting-edge algorithms. The proposed approach was found to be superior in both qualitative and quantitative analyses. Subjective evaluation showed improved visualization of retinal structures, while objective metrics such as contrast-to-noise ratio, segmentation accuracy, and edge preservation consistently outperformed benchmarks.

In retinal fundus imaging, the results show that the OAGF framework is effective in achieving both the enhancement and segmentation objectives. By providing clearer and more detailed images, this method facilitates more accurate diagnosis and monitoring of systemic diseases. Furthermore, its computational efficiency makes it suitable for integration into real-time diagnostic systems.

In conclusion, the Optimal Anisotropic Guided Filtering framework represents a significant advancement in retinal image analysis. Its capacity to improve image quality and precisely segment vascular structures resolves enduring challenges in the domain, facilitating the advancement of enhanced diagnostic techniques in ophthalmology and other fields.

This article emphasizes the capacity of advanced image processing methods to revolutionize medical imaging and facilitate early intervention in healthcare. Uneven illumination and low contrast affect pre-processing and vessel enhancement steps, leading to suboptimal outputs. In lower-quality images, artifacts such as spurious edges and disconnected vessels are evident, indicating room for improvement in refining postprocessing techniques.

This work heavily leverages the DRIVE and STARE datasets, which, although standard and expert-annotated, are limited in size and diversity. These data were acquired in controlled environments and possibly do not capture variability introduced by diverse fundus cameras, patient populations, or in vivo clinical environments. While OAGF performed well across diverse quality images, mild degradation in extreme low-contrast cases was seen. Future efforts will involve evaluation across larger, multi-institutional datasets in order to test for generalizability and further optimize the approach for wider clinical use.

Future work aims to integrate OAGF with deep learning models with datasets such as CHASE_DB1 and HRF, specifically convolutional neural networks (CNNs), to enhance vessel contrast and provide real-time clinical decision-support systems, combining interpretability and feature abstraction.

## Author contributions

**Conceptualization:** Samreen Fiza, K. Reddy Madhavi, Venkataramana Guntreddi.

**Formal analysis:** Subba Rao Polamuri.

**Investigation:** Subba Rao Polamuri.

**Methodology:** G. Tirumala Vasu, Venkataramana Guntreddi.

**Resources:** Thejaswini R.

**Supervision:** Venkataramana Guntreddi.

**Validation:** Venkataramana Guntreddi.

**Writing – original draft:** K. Reddy Madhavi, Venkataramana Guntreddi.

**Writing – review & editing:** Thejaswini R.

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
