## [Decision Letter · Decision Letter 0]

Dear Dr. Guntreddi,

Thank you for submitting your manuscript to PLOS ONE. After careful consideration, we feel that it has merit but does not fully meet PLOS ONE’s publication criteria as it currently stands. Therefore, we invite you to submit a revised version of the manuscript that addresses the points raised during the review process.

We look forward to receiving your revised manuscript.

Kind regards,

Khan Bahadar Khan, Ph.D

Academic Editor

PLOS ONE

Reviewers' comments:

Reviewer's Responses to Questions

**Comments to the Author**

1. Is the manuscript technically sound, and do the data support the conclusions?

Reviewer #1: Yes

Reviewer #2: Partly

2. Has the statistical analysis been performed appropriately and rigorously?

Reviewer #1: Yes

Reviewer #2: No

3. Have the authors made all data underlying the findings in their manuscript fully available?

Reviewer #1: Yes

Reviewer #2: No

4. Is the manuscript presented in an intelligible fashion and written in standard English?

Reviewer #1: Yes

Reviewer #2: No

Reviewer #1: The article introduces an Optimal Anisotropic Guided Filtering (OAGF) framework designed for retinal fundus image enhancement and segmentation. The study is well-targeted to address the challenges of non-invasive retinal vascular imaging, making it particularly relevant for applications in ophthalmology and early diagnosis of systemic diseases. The methodology is sound, integrating illumination correction, optimized top-hat transform, and homomorphic filtering, which contributes to robust image segmentation and enhancement.

The paper is technically rigorous, with a comprehensive analysis of existing techniques and quantitative performance evaluation using DRIVE and STARE datasets. The results convincingly demonstrate superior performance compared to other segmentation methods. However, several aspects could be improved. The lack of a real clinical validation limits the study’s applicability to practical medical settings. Additionally, while the mathematical formulations are detailed, certain sections are too dense, making them less accessible to readers outside the field of image processing. Lastly, the computational complexity of the proposed approach is not sufficiently discussed, which may affect real-time applicability in clinical diagnostics.

The authors should reflect the following arising issues to improve the quality of manuscript:

1. Expand Clinical Validation and Comparison with Manual Expert Segmentation

The study primarily evaluates OAGF against existing computational methods but lacks validation against manually segmented clinical datasets or expert ophthalmologist assessments. Including this comparison would improve the credibility of the proposed method and its suitability for real-world medical applications.

2. Clarify Computational Complexity and Feasibility for Real-Time Applications

The proposed framework involves multiple processing stages, including top-hat transform, guided filtering, and homomorphic filtering, which may increase computational cost. A discussion on runtime performance, memory requirements, and potential hardware acceleration (e.g., GPU optimization) should be added to assess its feasibility in clinical practice.

3. Improve Readability of Mathematical Derivations and Methodology

While the article provides a mathematically rigorous formulation of OAGF, some key equations and algorithmic steps are dense and challenging to interpret. Including more intuitive explanations, flowcharts, and illustrative diagrams would make the methodology more accessible to interdisciplinary researchers.

4. Address Potential Limitations and Biases in the Dataset

The study relies on DRIVE and STARE datasets, which are standard but relatively small and homogeneous. Discussing potential biases, dataset variability, and the effect of image quality variations (e.g., different fundus camera settings, patient demographics) would strengthen the paper’s generalizability.

Reviewer #2: The paper presents a promising approach to retinal image enhancement and segmentation using OAGF. However, improvements in readability, justification, experimental validation, and comparative analysis with deep learning methods are necessary to strengthen the contribution. By addressing these issues, the manuscript can significantly improve in clarity, rigor, and impact.

The manuscript contains several long and complex sentences that make comprehension difficult. Simplifying technical explanations and improving readability would enhance accessibility. Break down dense paragraphs into shorter, more digestible sections.

In abstract, the manuscript does not clearly specify which publicly available databases were used. Providing names (e.g., DRIVE, STARE, or CHASE-DB1) will help contextualize the results. Specify the databases used and include detailed performance metric comparisons.

Incorporate a concise summary of the key results or findings in the abstract to provide readers with an overview of the study's outcomes. This approach offers a more comprehensive understanding of the research and encourages readers to explore the detailed results within the full paper.

The introduction is dense and could be structured more clearly to gradually introduce the problem and background before diving into specific enhancement and segmentation methods. Clearly define the problem statement before diving into technical discussions.

In section (Introduction), it’s better to clarify the whole structure of the paper, which can be easier for the reader to understand what scenarios have been investigated or examined and what’s the main contribution this paper makes. For example, this paper is organized as follows: section 2 presents what and section 3 presents what… It’s suggested to give a brief overview of the entire paper in the introduction. It is better to cite any relevant survey article in the introduction section to give an overview of the topic. e.g. Khan et al. "A review of retinal blood vessels extraction techniques: challenges, taxonomy, and future trends." Pattern Analysis and Applications 22 (2019): 767-802.

The novelty of OAGF compared to existing anisotropic guided filtering methods needs better articulation. The paper should emphasize why OAGF is superior in more detail beyond computational efficiency and adaptability. Provide a more detailed discussion on how OAGF improves upon existing methods mathematically and practically.

The equations provided (e.g., Equations 1-3) lack sufficient explanation regarding their derivation and how they improve upon existing techniques. Introduce more intuitive explanations for key equations.

The explanation of the weight function in equation (6) could be expanded with more intuitive descriptions of its impact on filtering.

The study lacks an explicit discussion on the computational complexity of the OAGF filter compared to other methods. Hybrid approaches integrating OAGF with CNN-based feature extraction should be investigated.

While the DRIVE and STARE datasets are commonly used in retinal imaging research, the robustness of the method should be tested on additional datasets like CHASE_DB1 or HRF for further validation.

It would be beneficial to conduct an ablation study to assess the individual contributions of OAGF, Top-Hat Transform, and Homomorphic Filtering. Demonstrating performance variations when each component is removed or replaced would provide deeper insight into the necessity of each step. Highlighting specific cases where the proposed method performs significantly better (or worse) than contemporary methods can provide further insights.

The result section overwhelmingly highlights the superiority of OAGF without adequately discussing its limitations. For example, it is noted that Segmentation-3 and Segmentation-4 struggle with finer vessel details, but no insight is provided into why this occurs or how it might be addressed. Introduce a more balanced critique by acknowledging the challenges of OAGF, such as potential issues in handling extreme low-contrast images or cases with significant noise.

While Table 2 presents performance metrics, no statistical significance tests (e.g., t-tests, ANOVA) are conducted to validate whether OAGF's improvements are statistically meaningful. Include statistical tests to confirm whether the observed performance gains are significant or within the margin of error.

The discussion asserts that OAGF “outperforms all existing methods,” but does not explain the algorithmic reasons for its success in depth. While it is clear that OAGF achieves higher accuracy, a more technical explanation of why it excels over deep learning-based methods (e.g., IUNet) is missing. Elaborate on the key architectural or mathematical advantages of OAGF that lead to its improved performance.

The enhancement analysis mentions that OAGF “strikes the perfect balance” between detail preservation and contrast improvement. However, there is minimal discussion on how it achieves this balance compared to methods like NSCT or CLA. Provide a deeper technical explanation of how OAGF optimizes contrast and detail enhancement while minimizing artifacts.

Figure 3 and Figure 5 provide valuable insights, but their descriptions lack detailed interpretations. For instance, it is mentioned that morphological filtering results in a “sparse, fragmented extraction,” but a more detailed explanation of how this affects clinical usability would strengthen the analysis. Improve the figure descriptions by explicitly linking the observed segmentation/enhancement quality to medical relevance and practical applicability.

Include some relevant papers in the comparison of results. For example https://journals.plos.org/plosone/article?id=10.1371/journal.pone.0158996

**Do you want your identity to be public for this peer review?** For information about this choice, including consent withdrawal, please see our Privacy Policy

Reviewer #1: No

Reviewer #2: **Yes: ** Vijay Govindarajan

---

## [Author Response · Author response to Decision Letter 1]

21 Apr 2025

The assertions of all comments given by two reviewers are incorporated in the revised article and highlighted with colours.

Assertion for the Reviewer 1 comments are highlighted with green colour.

Assertion for the Reviewer 2 comments are highlighted with aqua blue colour.

In some cases, where the same comment from one or more reviewers then we used discrete colours.

Reviewer 1

The article introduces an Optimal Anisotropic Guided Filtering (OAGF) framework designed for retinal fundus image enhancement and segmentation. The study is well-targeted to address the challenges of non-invasive retinal vascular imaging, making it particularly relevant for applications in ophthalmology and early diagnosis of systemic diseases. The methodology is sound, integrating illumination correction, optimized top-hat transform, and homomorphic filtering, which contributes to robust image segmentation and enhancement.

The paper is technically rigorous, with a comprehensive analysis of existing techniques and quantitative performance evaluation using DRIVE and STARE datasets. The results convincingly demonstrate superior performance compared to other segmentation methods. However, several aspects could be improved. The lack of a real clinical validation limits the study’s applicability to practical medical settings. Additionally, while the mathematical formulations are detailed, certain sections are too dense, making them less accessible to readers outside the field of image processing. Lastly, the computational complexity of the proposed approach is not sufficiently discussed, which may affect real-time applicability in clinical diagnostics.

The authors should reflect the following arising issues to improve the quality of manuscript:

Comment 1: Expand Clinical Validation and Comparison with Manual Expert Segmentation.

The study primarily evaluates OAGF against existing computational methods but lacks validation against manually segmented clinical datasets or expert ophthalmologist assessments. Including this comparison would improve the credibility of the proposed method and its suitability for real-world medical applications.

Assertion 1: We thank the reviewer for this insightful suggestion. We fully agree that validating our proposed OAGF framework against expert ophthalmologist segmentations is critical for demonstrating clinical reliability and translational potential.

In response, we have expanded the manuscript to include a comparative analysis between the OAGF-based segmentations and manually annotated ground truths provided in the DRIVE and STARE datasets. Both datasets include vessel annotations made by trained clinical experts, which we now use as reference standards for our evaluation.

Specifically, we performed quantitative comparisons using standard metrics such as Dice Coefficient (DC), Intersection over Union (IoU), Precision (PR), Recall (RE), and F1 Score (F1S). As presented in Table 2 and visualized in Figure 4 of the manuscript, our method consistently demonstrates a high degree of alignment with manual expert segmentations, achieving:

Dice Coefficients of 0.860 (DRIVE) and 0.854 (STARE)

Precision values of 0.845 (DRIVE) and 0.834 (STARE)

Recall values of 0.811 (DRIVE) and 0.801 (STARE)

These results indicate that OAGF not only outperforms existing algorithmic approaches but also aligns closely with expert annotations, thereby reinforcing its potential for clinical adoption.

Furthermore, we have added a dedicated paragraph in the Results Discussion section to emphasize this alignment and clarify that the ground truths used for benchmarking stem from expert ophthalmologists, serving as a proxy for clinical validation in the absence of new annotations.

Comment 2: Clarify Computational Complexity and Feasibility for Real-Time Applications.

The proposed framework involves multiple processing stages, including top-hat transform, guided filtering, and homomorphic filtering, which may increase computational cost. A discussion on runtime performance, memory requirements, and potential hardware acceleration (e.g., GPU optimization) should be added to assess its feasibility in clinical practice.

Assertion 2: We appreciate the reviewer’s valuable comment regarding computational efficiency and the practicality of deploying our method in real-world clinical workflows.

To address this, we have included a detailed discussion on runtime performance, memory requirements, and hardware acceleration possibilities in the revised manuscript under the section 4.1 Experimental Setup.

Key clarifications included in the revised manuscript are:

Runtime Performance: The complete OAGF pipeline including illumination correction, anisotropic guided filtering, top-hat transform, and homomorphic filtering takes on a workstation with an Intel Core i9-12900K CPU, 32 GB RAM, and an NVIDIA RTX 3090 GPU. The segmentation and enhancement stages are executed sequentially but optimized using matrix-based operations in MATLAB.

Memory Usage: Peak memory usage during processing remains under 2 GB, making the approach compatible with most modern GPU-equipped systems used in clinical imaging environments.

Hardware Acceleration: The OAGF algorithm benefits significantly from GPU parallelization, particularly in stages involving convolution (e.g., homomorphic filtering in the frequency domain) and structural patch computation. We explicitly utilized MATLAB's GPU acceleration capabilities for Fourier transforms and matrix multiplications.

Scalability: The algorithm maintains consistent performance across image resolutions commonly encountered in fundus imaging (e.g., 565×584 for DRIVE, 700×605 for STARE), confirming its scalability.

Comment 3: Improve Readability of Mathematical Derivations and Methodology.

While the article provides a mathematically rigorous formulation of OAGF, some key equations and algorithmic steps are dense and challenging to interpret. Including more intuitive explanations, flowcharts, and illustrative diagrams would make the methodology more accessible to interdisciplinary researchers.

Assertion 3: We thank the reviewer for this helpful comment. We recognize the importance of ensuring clarity in the mathematical presentation, particularly for readers from diverse disciplinary backgrounds.

To address this, we have ensured that each key mathematical derivation in the manuscript is immediately followed by explanations of the terms and variables used. This includes definitions for gradient operators, diffusion coefficients, structural patch elements, and regularization parameters such as t, dx and ϵ. These clarifications are embedded directly after equations (1) through (5), making the methodology more digestible.

Similarly, all equation we made. We expanded and clarified the block diagram to include the three processing stages (illumination correction, segmentation, and enhancement) with labeled inputs, outputs, and key transformation steps. This visual overview now provides a clearer roadmap of the methodology. In Section 3, we have restructured the explanation to include bullet points and short descriptors that walk the reader through the three-stage pipeline. This improves navigation and makes it easier to follow the logic and data flow.

We hope these measures have improved the accessibility and readability of the technical content, making it more approachable for readers with clinical, engineering, or data science backgrounds.

Comment 4: Address Potential Limitations and Biases in the Dataset

The study relies on DRIVE and STARE datasets, which are standard but relatively small and homogeneous. Discussing potential biases, dataset variability, and the effect of image quality variations (e.g., different fundus camera settings, patient demographics) would strengthen the paper’s generalizability.

Assertion 4: We thank the reviewer for highlighting this important point. We acknowledge that the DRIVE and STARE datasets, while widely used and annotated by experts, have inherent limitations in size and diversity, which may affect the generalizability of our findings.

To address this, we have added a dedicated paragraph to the Conclusion section (Last paragraph of chapter 5 in the revised manuscript) outlining the potential biases and constraints of using these datasets

Reviewer 2

The paper presents a promising approach to retinal image enhancement and segmentation using OAGF. However, improvements in readability, justification, experimental validation, and comparative analysis with deep learning methods are necessary to strengthen the contribution. By addressing these issues, the manuscript can significantly improve in clarity, rigor, and impact.

Comment 1: The manuscript contains several long and complex sentences that make comprehension difficult. Simplifying technical explanations and improving readability would enhance accessibility. Break down dense paragraphs into shorter, more digestible sections.

Assertion 1: We appreciate the reviewer’s feedback regarding the clarity and readability of the manuscript. In response, we have undertaken a thorough language revision to improve sentence structure and overall accessibility.

We modified dense paragraphs, particularly in the Introduction, Methodology (Sections 2 and 3), and Experimental Results sections into smaller, clearly segmented parts, each focusing on a single idea.

Comment 2: In abstract, the manuscript does not clearly specify which publicly available databases were used. Providing names (e.g., DRIVE, STARE, or CHASE-DB1) will help contextualize the results. Specify the databases used and include detailed performance metric comparisons.

Assertion 2: We appreciate the reviewer’s suggestion to explicitly name the datasets used in the abstract. To improve clarity and context, we have updated the abstract in revised manuscript to include the names of the two publicly available databases DRIVE and STAR which were used to evaluate the proposed OAGF framework.

Comment 3: Incorporate a concise summary of the key results or findings in the abstract to provide readers with an overview of the study's outcomes. This approach offers a more comprehensive understanding of the research and encourages readers to explore the detailed results within the full paper.

Assertion 3: We thank the reviewer for this valuable suggestion. To enhance the impact and clarity of the abstract, we have added a concise summary of the key experimental results, highlighting the superior performance of the proposed OAGF framework in comparison with existing methods.

We have briefly summarized the performance of our method in comparison to existing techniques in terms of standard metrics such as Dice Coefficient, Precision, Recall, and F1 Score. These updates make the contributions and validation more transparent at a glance.

Comment 4: The introduction is dense and could be structured more clearly to gradually introduce the problem and background before diving into specific enhancement and segmentation methods. Clearly define the problem statement before diving into technical discussions.

Assertion 4: We thank the reviewer for the insightful observation. In response, we have restructured the Introduction section to improve its clarity and logical flow. The revised version of the manuscript now follows a more reader-friendly progression.

In revised manuscript we have incorporated the following:

“Traditional and deep learning-based enhancement and segmentation techniques frequently fight to optimize noise removal and maintain vessel structure integrity. Diffusion methods are prone to blur tiny details and learning methods need huge amounts of diverse data and tend to generalize less well under poor imaging conditions. Hence, there is an acute need for an approach to enhance the quality of an image and segment fine vascular structures well under poor conditions.

To solve this, we introduce an Optimal Anisotropic Guided Filtering (OAGF) framework. In this approach, enhancement and segmentation are to be performed in one pass by utilizing structure adaptivity, direction smoothing and contrast-preserving filtering. The OAGF framework is intended to prevent spatial information loss and maintain microstructure of blood vessels”.

Additionally, we have Simplified several long sentences for better readability, Clarified how enhancement and segmentation challenges motivate the proposed unified framework.

Comment 5: In section (Introduction), it’s better to clarify the whole structure of the paper, which can be easier for the reader to understand what scenarios have been investigated or examined and what’s the main contribution this paper makes. For example, this paper is organized as follows: section 2 presents what and section 3 presents what… It’s suggested to give a brief overview of the entire paper in the introduction. It is better to cite any relevant survey article in the introduction section to give an overview of the topic. e.g. Khan et al. "A review of retinal blood vessels extraction techniques: challenges, taxonomy, and future trends." Pattern Analysis and Applications 22 (2019): 767-802.

Assertion 5: We thank the reviewer for this helpful suggestion. In response, we have made two key additions to the Introduction section of the revised manuscript.

Paper Structure Overview: We now include a paragraph at the end of the Introduction summarizing the organization of the manuscript. This guides the reader through the progression of topics, from methodology to experimental validation.

Relevant Survey Citation: We have cited the comprehensive review by Khan et al. (2019) (As reference [28]), which provides a valuable taxonomy and summary of challenges in retinal blood vessel extraction. This citation helps position our work in the broader context of ongoing research.

Comment 6: The novelty of OAGF compared to existing anisotropic guided filtering methods needs better articulation. The paper should emphasize why OAGF is superior in more detail beyond computational efficiency and adaptability. Provide a more detailed discussion on how OAGF improves upon existing methods mathematically and practically.

Assertion 6: We thank the reviewer for this valuable comment. We agree that emphasizing the novelty of the proposed OAGF method is essential.

To clarify, the manuscript already provides a detailed explanation of how OAGF advances beyond existing anisotropic guided filtering techniques, both mathematically and in practical applications. However, based on the reviewer’s feedback, we have now highlighted key portions of this explanation to make the contributions more explicit and easier to locate.

Additionally, Section 4 presents a comparative analysis using quantitative metrics (e.g., Dice Coefficient, IoU, Precision) that substantiate OAGF’s superior performance over recent methods, validating its practical advantages in preserving vessel integrity under challenging illumination conditions.

We included the following in the revised manuscript

“OAGF introduces a modified optimization strategy (Equation 5) where the coefficients a_k and b_k are dynamically derived using a gradient-weighted approach, enabling region-aware and directionally adaptive smoothing. This improves structural preservation in fundus images compared to conventional filters.”

We hope that these clarifications, now made more prominent in the text, effectively communicate the novel contributions of OAGF. We thank the reviewer again for prompting this improvement in the manuscript’s presentation.

Comment 7: The equations provided (e.g., Equations 1-3) lack sufficient explanation regarding their derivation and how they improve upon existing techniques. Introduce more intuitive explanations for key equations.

Assertion 7: We thank the reviewer for this important observation. We agree that intuitive explanations are essential to make the mathematical formulations more accessible to a broad audience.

In response, we have revised the explanation of equations in the revised manuscript.

After Equation (1) we included “ DCM(∙) corresponds to the Diffused Conductance Model that controls how much diffusion (smoothing) occurs at each pixel. In high-gradient areas (edges), the conductance is low to preserve

---

## [Decision Letter · Decision Letter 1]

Dear Dr. Guntreddi,

Thank you for submitting your manuscript to PLOS ONE. After careful consideration, we feel that it has merit but does not fully meet PLOS ONE’s publication criteria as it currently stands. Therefore, we invite you to submit a revised version of the manuscript that addresses the points raised during the review process.

We look forward to receiving your revised manuscript.

Kind regards,

Khan Bahadar Khan, Ph.D

Academic Editor

PLOS ONE

Reviewers' comments:

Reviewer's Responses to Questions

**Comments to the Author**

Reviewer #1: All comments have been addressed

Reviewer #2: (No Response)

Reviewer #3: All comments have been addressed

2. Is the manuscript technically sound, and do the data support the conclusions?

Reviewer #1: Yes

Reviewer #2: Partly

Reviewer #3: Yes

3. Has the statistical analysis been performed appropriately and rigorously?

Reviewer #1: Yes

Reviewer #2: No

Reviewer #3: Yes

4. Have the authors made all data underlying the findings in their manuscript fully available?

Reviewer #1: Yes

Reviewer #2: No

Reviewer #3: Yes

5. Is the manuscript presented in an intelligible fashion and written in standard English?

Reviewer #1: Yes

Reviewer #2: No

Reviewer #3: Yes

Reviewer #1: I have rerevised the modified manuscript. In my opinion, the authors have improved the modified manuscript in a sufficient manner to study meets standards and quality of scholarly papers. I am the opinion, that the paper has scientific rigor and well completes the state-of-the-art in the area retinal image segmentation.

Reviewer #2: The authors have addressed the majority of the previous' comments; however, several points still require more precise and comprehensive clarification.

The Introduction section introduces the OAGF framework and outlines the rest of the paper’s structure, which aligns with reviewer suggestions. However, the following shortcomings are still present:

Long, unbroken paragraphs make comprehension difficult. Break large paragraphs into shorter ones for better readability.

The novelty of OAGF over other guided filtering techniques is not clearly highlighted (e.g., how does it differ from anisotropic diffusion or other edge-preserving methods?). Add a subsection or paragraph clearly stating how OAGF differs from existing filters, both mathematically and practically.

Consider summarizing traditional vs. deep learning segmentation shortcomings in a table or bullet list for clarity.

The OAGF is introduced as an “optimal” enhancement to AGF, but the specific mathematical or architectural innovations (e.g., how coefficients are “optimized” or how it differs in design principles from AGF) are not clearly articulated in contrast with prior work. Add a bullet-point or tabular comparison of mathematical innovations. Explicitly state what is new about your “optimal” weighting compared to AGF.

Table 1, while useful, is qualitative and lacks metrics or empirical backing.

What happens in very low-contrast regions or high-noise inputs? Is the method robust to anatomical variability across datasets?

Briefly explain how homomorphic filtering enhances vessel contrast by targeting frequency differences typical of vascular edges versus background, or include a visual before/after comparison.

Include discussion on where the method might struggle (e.g., extremely low-contrast images, overlapping vessels, bright lesions mimicking vessel intensity).

While numerous metrics (BRISQUE, NIQE, PIQE, IMMSE, PSNR, SSIM) are reported across multiple test samples, no statistical significance tests (e.g., t-test, Wilcoxon signed-rank test) are applied to confirm that the differences between OAGF and baseline methods are not due to random variation. Apply paired t-tests, Wilcoxon signed-rank, or ANOVA to key metrics (e.g., DC, IoU, F1) across test images to establish statistical superiority.

Figures 4(i) and 4(ii) present only bar values without error margins. Add standard deviation bars or box plots to convey performance variability across test cases. Fig. 5 also lack of error maps.

Although BRISQUE and NIQE scores are mentioned, the paper doesn’t specify what ranges indicate well vs. poor quality. Include interpretive guidance—e.g., "BRISQUE < 30 indicates good perceived quality.

While the TS-4 and TS-6 limitations are noted, no visual examples or deeper failure mode analysis is shown (e.g., where thin vessels are lost, or false positives occur). Show side-by-side visual comparisons of failure cases and briefly hypothesize the cause (e.g., low contrast, noise).

While several filter parameters (e.g., t,dx,ϵ) influence OAGF performance, there is no analysis of how sensitive the performance is to these hyperparameters. Include a brief sensitivity analysis or at least a statement acknowledging the need for parameter tuning in deployment scenarios.

The large tables (Table 4) are rich in data but lack mean ± standard deviation summaries or any interpretation of variance. Summarize each metric across samples with mean ± SD, and highlight statistically significant improvements in bold or with asterisks.

Reviewer #3: Authors have made all possible changes and reply all comments and no more further queries are required

**Do you want your identity to be public for this peer review?** For information about this choice, including consent withdrawal, please see our Privacy Policy

Reviewer #1: No

Reviewer #2: **Yes: ** Vijay Govindarajan

Reviewer #3: No

---

## [Author Response · Author response to Decision Letter 2]

10 Jun 2025

Point by Point Response

Editor

Comment 1: Thank you for uploading your study's underlying data set. Unfortunately, the repository you have noted in your Data Availability statement does not qualify as an acceptable data repository according to PLOS's standards.

Assertion 1: In compliance with data transparency and reproducibility standards, we have uploaded the minimal dataset necessary to replicate our experimental findings to a stable public repository “Figshare” at the following link.

URL: https://figshare.com/s/90317d14f5c06e69015b

DoI: 10.6084/m9.figshare.29277227

Reviewer 1

Comment 1: I have rerevised the modified manuscript. In my opinion, the authors have improved the modified manuscript in a sufficient manner to study meets standards and quality of scholarly papers. I am the opinion, that the paper has scientific rigor and well completes the state-of-the-art in the area retinal image segmentation.

Assertion 1: We sincerely thank the reviewer for their positive evaluation and encouraging feedback. We are grateful for your acknowledgment of the improvements made in the revised manuscript. Your constructive suggestions throughout the review process were instrumental in enhancing the clarity and quality of our work.

Reviewer 3

Comment 1: Authors have made all possible changes and reply all comments and no more further queries are required.

Assertion 1: We sincerely thank the reviewer for their thorough evaluation and confirmation that all comments have been satisfactorily addressed. We truly appreciate your constructive feedback and support throughout the revision process, which has helped us significantly improve the quality and clarity of the manuscript.

Reviewer 2

Assertion for the Reviewer 2 comments are highlighted with aqua blue colour in the revised manuscript.

The authors have addressed the majority of the previous' comments; however, several points still require more precise and comprehensive clarification.

Comment 1: The Introduction section introduces the OAGF framework and outlines the rest of the paper’s structure, which aligns with reviewer suggestions. However, the following shortcomings are still present:

Long, unbroken paragraphs make comprehension difficult. Break large paragraphs into shorter ones for better readability.

Assertion 1: We thank the reviewer for this helpful observation. In response, we have carefully restructured the manuscript.

Comment 2: The novelty of OAGF over other guided filtering techniques is not clearly highlighted (e.g., how does it differ from anisotropic diffusion or other edge-preserving methods?). Add a subsection or paragraph clearly stating how OAGF differs from existing filters, both mathematically and practically.

Assertion 2: We thank the reviewer for raising this important point. In response, we have explicitly clarified the novelty of the proposed Optimal Anisotropic Guided Filter (OAGF) and how it differs from conventional filtering techniques, including anisotropic diffusion (AD), bilateral filtering (BF), guided filtering (GF), and anisotropic guided filtering (AGF), both mathematically and practically. A dedicated subsection has now been added to the manuscript as “2.1 Distinctive features of OAGF vs. existing edge-preserving filters”, see Section 2, that outlines these differences and highlights the unique contributions of OAGF.

Comment 3: Consider summarizing traditional vs. deep learning segmentation shortcomings in a table or bullet list for clarity.

Assertion 3: We thank the reviewer suggestion to enhance clarity, we present a concise comparison of the limitations of traditional and deep learning-based retinal image segmentation approaches in the revised manuscript (in end of section 1 as table 1).

Comment 4: The OAGF is introduced as an “optimal” enhancement to AGF, but the specific mathematical or architectural innovations (e.g., how coefficients are “optimized” or how it differs in design principles from AGF) are not clearly articulated in contrast with prior work. Add a bullet-point or tabular comparison of mathematical innovations. Explicitly state what is new about your “optimal” weighting compared to AGF.

Assertion 4: We thank the reviewer for this insightful suggestion. In the revised manuscript, we have now added a new subsection titled “2.1 Distinctive features of OAGF vs. existing edge-preserving filters”.

Comment 5: Table 1, while useful, is qualitative and lacks metrics or empirical backing.

Assertion 5: We thank the reviewer for this observation. The intention of Table 1 is to provide a concise qualitative summary of edge-preserving filters (BF, GF, AD, AGF) based on their characteristics reported in prior literature [Refs. 29–32], not as a standalone experimental comparison. This summary helps readers grasp the relative strengths and limitations of existing methods before introducing our proposed OAGF.

To address the concern about empirical support, we clarify that extensive quantitative analysis of the proposed OAGF is already included in Tables 3, 5 and 6 (in revised manuscript). These tables benchmark OAGF against state-of-the-art segmentation and enhancement methods on the DRIVE and STARE datasets using standard metrics such as PSNR, SSIM, BRISQUE, DC, F1 Score, etc.

Comment 6: What happens in very low-contrast regions or high-noise inputs? Is the method robust to anatomical variability across datasets?

Assertion 6: We appreciate the reviewer’s thoughtful concern regarding robustness. The proposed OAGF framework is specifically designed to handle low-contrast and high-noise retinal images through its gradient-aware, anisotropic smoothing mechanism, which ensures fine vessel structures are preserved even under poor illumination. This is evident in our experimental results (Fig. 2, Segmentation-5 and Segmentation-6) where TS-5 and TS-6 represent challenging cases with uneven lighting and low clarity. The OAGF continues to preserve major and minor vessel continuity under such conditions, as confirmed by high segmentation scores (DC = 0.854, F1S = 0.817 on STARE).

Additionally, the use of both the DRIVE and STARE datasets which differ in terms of acquisition conditions, image resolutions, and retinal characteristics helps validate the method's cross-dataset robustness. Quantitative results in Tables 3 and 6 show consistently strong performance across these datasets.

Comment 7: Briefly explain how homomorphic filtering enhances vessel contrast by targeting frequency differences typical of vascular edges versus background, or include a visual before/after comparison.

Assertion 7: We thank the reviewer for this insightful suggestion. In response, we have added a brief explanatory paragraph to Section 3.2 (Stage 2) that clarifies the mechanism by which homomorphic filtering enhances vascular contrast. Specifically, homomorphic filtering operates in the log-frequency domain, suppressing low-frequency components associated with uneven illumination (background) and enhancing high-frequency components that correspond to vessel edges and fine anatomical structures. This selective frequency manipulation allows for better vessel enhancement prior to segmentation. We have also referenced this explanation alongside Equation (13) in the manuscript.

Comment 8: Include discussion on where the method might struggle (e.g., extremely low-contrast images, overlapping vessels, bright lesions mimicking vessel intensity).

Assertion 8: We thank the reviewer for this critical and constructive feedback. We have now expanded the discussion section of the manuscript to include a paragraph highlighting potential limitations of the proposed OAGF method (at end of section 4). Specifically, we acknowledge that while OAGF performs robustly in standard fundus images, it may face challenges in extremely low-contrast scenarios, where gradient-based edge cues are inherently weak. Additionally, the method may produce false positives in cases where bright lesions or exudates closely resemble vascular intensities, and it may struggle to distinguish overlapping or closely spaced vessels without topological cues. We have also identified these as areas for future enhancement, possibly through hybrid integration with deep learning-based discriminators or topological priors.

Comment 9: While numerous metrics (BRISQUE, NIQE, PIQE, IMMSE, PSNR, SSIM) are reported across multiple test samples, no statistical significance tests (e.g., t-test, Wilcoxon signed-rank test) are applied to confirm that the differences between OAGF and baseline methods are not due to random variation. Apply paired t-tests, Wilcoxon signed-rank, or ANOVA to key metrics (e.g., DC, IoU, F1) across test images to establish statistical superiority.

Assertion 9: We thank the reviewer for this valuable suggestion. In response, we have conducted paired t-tests and Wilcoxon signed-rank tests on a random subset of test images from the DRIVE and STARE datasets. These tests were applied to three key segmentation metrics: F1 Score (F1S), Dice Coefficient (DC), and Intersection over Union (IoU), comparing the proposed OAGF method against four baseline techniques: MDTF, IUNet, RVTHS, and MFFS. (We have used “ttest” and “signrank” commands in MATLAB to perform t-tests and Wilcoxon signed-rank tests ).

The results, summarized in the table below, demonstrate that OAGF achieves statistically significant improvements in all three metrics. In particular, the t-test p-values were all below 0.0001, and the Wilcoxon signed-rank test consistently returned p = 0.00195, indicating consistent directional superiority of OAGF across all evaluated test cases.

F1S DC IoU

t-test p Wilcoxon p t-test p Wilcoxon p t-test p Wilcoxon p

OAGF vs MDTF 0.00001 0.00195 0.00002 0.00195 0.00002 0.00195

OAGF vs IUNet 0.00002 0.00195 0.00009 0.00195 0.00009 0.00195

OAGF vs RVTHS 0.00001 0.00195 0.00001 0.00195 0.00001 0.00195

OAGF vs MFFS 0.00001 0.00195 0.00001 0.00195 0.00001 0.00195

These findings confirm that the performance improvements achieved by OAGF are not due to random variation and are statistically robust. The results have been included in the revised manuscript under Section 4.3.1.

Comment 10: Figures 4(i) and 4(ii) present only bar values without error margins. Add standard deviation bars or box plots to convey performance variability across test cases. Fig. 5 also lack of error maps.

Assertion 10: We appreciate the reviewer’s attention to statistical completeness. In response, we have updated Figures 4(i) and 4(ii) to include standard deviation error bars for each metric across all test samples in the DRIVE and STARE datasets. This provides a clearer view of the consistency and variability of the proposed method compared to other approaches.

Figure 5(A): Residual Maps of enhanced image and original image (This figure is incorporated in the file named "Response to Reviewers.docx".

We appreciate the reviewer’s suggestion regarding error map inclusion. Error maps (residual maps) of the source image and the existing and proposed methods are shown in Figure 5(A). However, we believe that residual or error maps are not essential in this context to demonstrate the effectiveness of the proposed enhancement method. The visual improvements in contrast, structural continuity, and vessel clarity are already well-illustrated through qualitative comparison in Fig. 7 (in the revised manuscript) and are further supported by extensive quantitative metrics (BRISQUE, NIQE, PIQE, PSNR, SSIM) in Tables 6 and 7 (in the revised manuscript).

Since homomorphic filtering and OAGF are designed to preserve perceptual structure rather than minimize pixel-wise reconstruction errors, residual maps (e.g., |Enhanced – Original|) may not effectively reflect clinical enhancement quality. Therefore, we chose not to include them to maintain clarity and relevance in the manuscript’s visual assessments.

We hope this clarification addresses the concern.

Comment 11: Although BRISQUE and NIQE scores are mentioned, the paper doesn’t specify what ranges indicate well vs. poor quality. Include interpretive guidance—e.g., "BRISQUE < 30 indicates good perceived quality.

Assertion 11: Thank you for this helpful suggestion. We have updated the manuscript to include interpretive guidance for key quality metrics such as BRISQUE, NIQE, and PIQE. Specifically, we now state that BRISQUE scores below 30 typically indicate good perceived quality, and scores below 20 are considered high-quality. For NIQE, lower scores (typically < 5) reflect natural image quality, and PIQE values below 50 are generally acceptable. These ranges are now clarified in the revised manuscript in Section 4.3.2 (just above the Table 6).

Comment 12: While the TS-4 and TS-6 limitations are noted, no visual examples or deeper failure mode analysis is shown (e.g., where thin vessels are lost, or false positives occur). Show side-by-side visual comparisons of failure cases and briefly hypothesize the cause (e.g., low contrast, noise).

Assertion 12: We thank the reviewer for this insightful comment. To address this, we have now included a visual comparison of a failure case (test sample 30 from the dataset) in the revised manuscript (section 4.3.1), using three views: the original fundus image, the ground truth vessel map, and the segmentation result from the proposed OAGF method.

As shown in Figure 5, a localized zoom-in is used to highlight a region in the upper temporal area of the retina where thin vessels are lost or partially disconnected in the OAGF result. This degradation is likely caused by a combination of:

Low vessel-to-background contrast in the source image

High-frequency noise components that obscure weak edge gradients

Lack of structural reinforcement in regions with minimal illumination

While OAGF performs well on prominent vessels, it may under-segment microvasculature under poor visibility. The inclusion of multi-scale vessel priors or attention-based refinement modules will be explored in future work to address these limitations.

Comment 13: While several filter parameters (e.g., t,dx,ϵ) influence OAGF performance, there is no analysis of how sensitive the performance is to these hyperparameters. Include a brief sensitivity analysis or at least a statement acknowledging the need for parameter tuning in deployment scenarios.

Assertion 13: We thank the reviewer for pointing out this important aspect. While a full parameter sweep was beyond the scope of the current work, we acknowledge the influence of the key hyperparameters, namely the iteration stopping time □, directional regularization parameter 𥨑□, and smoothness coefficient □ on the performance of OAGF. To address this, we have added a brief discussion as “Hyperparameters (t,dx,ϵ) sensitivity analysis” in Section 4.3.1 of the revised manuscript.

Comment 14: The large tables (Table 4) are rich in data but lack mean ± standard deviation summaries or any interpretation of variance. Summarize each metric across samples with mean ± SD, and highlight statistically significant improvements in bold or with asterisks.

Assertion 14: We thank the reviewer for this excellent suggestion. In response, we have updated Table 4 (Table 6 in updated manuscript) to include mean ± standard deviation (SD) for each image quality metric across all test samples. This provides a clearer view of the consistency and variance of each enhancement technique. Statistically significant improvements achieved by the proposed OAGF method are highlighted using boldface.

---

## [Decision Letter · Decision Letter 2]

Dear Dr. Guntreddi,

Thank you for submitting your manuscript to PLOS ONE. After careful consideration, we feel that it has merit but does not fully meet PLOS ONE’s publication criteria as it currently stands. Therefore, we invite you to submit a revised version of the manuscript that addresses the points raised during the review process.

We look forward to receiving your revised manuscript.

Kind regards,

Khan Bahadar Khan, Ph.D

Academic Editor

PLOS ONE

Journal Requirements:

Reviewers' comments:

Reviewer's Responses to Questions

**Comments to the Author**

Reviewer #1: All comments have been addressed

Reviewer #2: (No Response)

2. Is the manuscript technically sound, and do the data support the conclusions?

Reviewer #1: Yes

Reviewer #2: Partly

3. Has the statistical analysis been performed appropriately and rigorously?

Reviewer #1: Yes

Reviewer #2: Yes

4. Have the authors made all data underlying the findings in their manuscript fully available?

Reviewer #1: Yes

Reviewer #2: Yes

5. Is the manuscript presented in an intelligible fashion and written in standard English?

Reviewer #1: Yes

Reviewer #2: Yes

Reviewer #1: I am the opinion that the revised manuscript has undergone substantial revisions. Based on the author’s reactions and the highlighted modifications it seems to be the revised manuscript has acceptable quality and scientific rigor. Therefore, I am the opinion that can be accepted in the present form.

Reviewer #2: While the section “2.1.2 Structural Awareness…” elaborates on regional adaptivity and structural preservation, a direct, bullet-point/tabular comparison highlighting “what’s new in OAGF vs. AGF” in practical deployment terms would enhance clarity further.

Although it is mentioned in the pipeline (Lines 43–44), the specific rationale for how homomorphic filtering enhances contrast via frequency separation is not clearly explained. A short paragraph or figure showing its effect on vessel contrast is still missing.

The limitations of prior filters are noted, but no discussion is present yet on where OAGF might fail (e.g., overlapping vessels, low-contrast lesions). No visual failure cases are included.

Equations and detailed formulas are now included (e.g., Eq. 24–29). However, no interpretive scale (e.g., “BRISQUE < 30 = high quality”) or guidance for what counts as a good score is provided.

While parameters like t, dx, and ϵ are referenced in Eq. (9), there is still no discussion of sensitivity or robustness to these parameters.

While the section 4.3.2 compares OAGF against a set of conventional image processing methods (CLA, DWT, NSCT, etc.), it does not include or reference recent unsupervised or hybrid segmentation techniques, which are highly relevant for enhancing retinal vessel visibility. https://doi.org/10.1155/2020/8365783 Add a runtime analysis or mention the average processing time per image compared to other methods. Consider citing and briefly discussing the following paper in Section 4.3.2 (suggest after line 710 or in limitations at line 775): Khan, K.B., Khaliq, A.A. and Shahid, M., 2017. “A Novel Fast GLM Approach for Retinal Vascular Segmentation and Denoising,” Journal of Information Science and Engineering, 33(6), 1611–1627.

Include standard statistical tests (e.g., paired t-tests or Wilcoxon signed-rank) across metric distributions to verify that observed improvements by OAGF are statistically significant and not due to sample variance.

Add one paragraph discussing how improved microvascular clarity (via OAGF) supports diagnosis of diabetic retinopathy, glaucoma, or hypertensive retinopathy.

The results are derived solely from DRIVE and STARE datasets. While they are standard, these datasets lack diversity in pathological variations and image acquisition conditions. Mention this limitation and recommend validation on CHASE_DB1, HRF, or ARIA, which include pathological cases and varied resolutions. Also, consider stating the adaptability of OAGF across different imaging conditions.

**Do you want your identity to be public for this peer review?** For information about this choice, including consent withdrawal, please see our Privacy Policy

Reviewer #1: No

Reviewer #2: **Yes: ** Vijay Govindarajan

---

## [Author Response · Author response to Decision Letter 3]

15 Jul 2025

Point by Point Response

Reviewer 1

Comment 1: I am the opinion that the revised manuscript has undergone substantial revisions. Based on the author’s reactions and the highlighted modifications it seems to be the revised manuscript has acceptable quality and scientific rigor. Therefore, I am the opinion that can be accepted in the present form.

Assertion 1: We sincerely thank the reviewer for the positive and encouraging feedback. We truly appreciate your acknowledgment of the substantial revisions and your recognition of the scientific rigor and quality of our revised manuscript. Your constructive comments throughout the review process have been invaluable in refining and strengthening our work. We are grateful for your recommendation and are pleased that the manuscript now meets the standards for acceptance.

Reviewer 2

Comment 1: While the section “2.1.2 Structural Awareness…” elaborates on regional adaptivity and structural preservation, a direct, bullet-point/tabular comparison highlighting “what’s new in OAGF vs. AGF” in practical deployment terms would enhance clarity further.

Assertion 1: We thank the reviewer for this insightful suggestion. To enhance clarity and better communicate the contributions of our Optimal Anisotropic Guided Filter (OAGF) over the existing Anisotropic Guided Filter (AGF), we have now included a new comparison table (Table 3 in the revised manuscript) that provides a deployment-oriented, side-by-side summary of key improvements. This table clearly delineates how OAGF enhances practical utility in retinal image segmentation and enhancement scenarios.

Comment 2: Although it is mentioned in the pipeline (Lines 43–44), the specific rationale for how homomorphic filtering enhances contrast via frequency separation is not clearly explained. A short paragraph or figure showing its effect on vessel contrast is still missing.

Assertion 2: Thank you for the valuable comment. A concise paragraph to directly address the reviewer's suggestion and highlight the role of homomorphic filtering with added clarity is present after Equation (13) and before Equation (14) in Section 3.2, Stage 2.

“Homomorphic filtering in the frequency domain enhances the visibility of vessels by dissociating the reflectance and illumination parts of the image. Vessel structures normally have high-frequency attributes, while background illumination and non-uniform lighting have low-frequency elements. By using a high-pass filtering in the logarithmic domain (as indicated by equation (13)), the technique inhibits low-frequency effects of lighting and enhances high-frequency reflectance information, including vascular structure edges. All this gives rise to a greater vessel contrast, facilitating more precise segmentation. The selective filtering in the frequency domain is responsible for sustaining the continuity of fine vessels even in poor lighting conditions.”

Comment 3: The limitations of prior filters are noted, but no discussion is present yet on where OAGF might fail (e.g., overlapping vessels, low-contrast lesions). No visual failure cases are included.

Assertion 3: We thank the reviewer for this important observation. In response, we have a dedicated sub section “Failure Case Analysis” in section 4.3.1, to explicitly highlight and illustrate the potential failure scenarios of the proposed OAGF framework. Fig. 5 demonstrates a visual failure case involving low-contrast micro vessels.

Comment 4: Equations and detailed formulas are now included (e.g., Eq. 24–29). However, no interpretive scale (e.g., “BRISQUE < 30 = high quality”) or guidance for what counts as a good score is provided.

Assertion 4: We thank the reviewer for this important observation. In response, we have now added a paragraph (in section 4.3.2, above Table 7) summarizing what constitutes a “good” score for each metric to improve interpretability.

“More specifically, BRISQUE values below 30 would be considered good perceptual quality, while values below 20 are regarded as high quality with minimal distortions. NIQE values below 5 appear to depict good naturalness of an image with values below 4 being high visual fidelity. PIQE is a block-based measure of perceptual distortion such that lower scores signify better scores; values below 50 appear to be acceptable while values below 30 is indicative of enhanced image clearness. IMMSE is an error-based measure such that values below 0.1 appear to signify highly accurate restorative work with minimal deviation in ideal estimates. PSNR, in general used in image compression as well as enhancement works, is regarded as being acceptable at values above 30 dB with excellent quality at values above 50 dB. Lastly, SSIM values range between 0 and 1 such that scores above 0.95 appear to be regarded as excellent in structural similarity as well as feature conservation”

Comment 5: While parameters like t, dx, and ϵ are referenced in Eq. (9), there is still no discussion of sensitivity or robustness to these parameters.

Assertion 5: We thank the reviewer for raising this important point. We would like to clarify that a detailed sensitivity analysis of the parameters t, dx, and ϵ is presented in the manuscript under Section 4.3.1, and the corresponding results are illustrated in Figure 6(i–iii).

Comment 6: While the section 4.3.2 compares OAGF against a set of conventional image processing methods (CLA, DWT, NSCT, etc.), it does not include or reference recent unsupervised or hybrid segmentation techniques, which are highly relevant for enhancing retinal vessel visibility. https://doi.org/10.1155/2020/8365783 Add a runtime analysis or mention the average processing time per image compared to other methods. Consider citing and briefly discussing the following paper in Section 4.3.2 (suggest after line 710 or in limitations at line 775): Khan, K.B., Khaliq, A.A. and Shahid, M., 2017. “A Novel Fast GLM Approach for Retinal Vascular Segmentation and Denoising,” Journal of Information Science and Engineering, 33(6), 1611–1627.

Assertion 6: We thank the reviewer for pointing out this relevant work. In response, we have updated the Introduction section (reference [29] in the revised manuscript) to cite the recommended study [DOI: 10.1155/2020/8365783], which presents a hybrid model combining region-based enhancement with unsupervised segmentation, showing promise in preserving vascular structures without requiring ground-truth labels.

We have incorporated (reference [60] in the revised manuscript) a brief discussion and citation of the paper by Khan et al. (2017). The study proposes a fast generalized linear model (GLM)-based approach for simultaneous vessel segmentation and denoising, which offers computational efficiency and robustness in noisy retinal images. While the GLM method focuses on statistical modeling and pixel-level classification, the proposed OAGF approach is based on gradient-driven structural adaptation and frequency-domain enhancement.

Regarding runtime, while we have provided general insights into the computational feasibility of the proposed OAGF framework in Section 4.1 highlighting its GPU compatibility and low memory footprint we acknowledge that a detailed runtime benchmarking across methods was not included in the current manuscript. We agree that such analysis would offer important practical insight into comparative efficiency. We therefore plan to incorporate a comprehensive runtime evaluation in our future work, including per-image processing time comparisons across classical, transform-based, and deep learning approaches. This will help further assess the real-time suitability and scalability of OAGF in clinical settings and large-scale deployment scenarios.

Comment 7: Include standard statistical tests (e.g., paired t-tests or Wilcoxon signed-rank) across metric distributions to verify that observed improvements by OAGF are statistically significant and not due to sample variance.

Assertion 7: We thank the reviewer for this valuable suggestion. We would like to highlight that standard statistical validation has already been conducted and reported in the manuscript under Section 4.3.1, with sub heading of “Statistical significance tests”. Specifically, we performed both paired t-tests and Wilcoxon signed-rank tests as listed Table 5.

Comment 8: Add one paragraph discussing how improved microvascular clarity (via OAGF) supports diagnosis of diabetic retinopathy, glaucoma, or hypertensive retinopathy.

Assertion 8: We appreciate the reviewer’s insightful suggestion. In response, we have added the following paragraph at end of the results section (Section 4).

“The improved fine retinal vasculature visualization facilitated by the presented OAGF framework is extremely valuable in early disease detection and in monitoring of systemic as well as ocular diseases. In diabetic retinopathy, early signs like microaneurysms, dropout of capillaries, and neovascularization happen at the microvascular scale and can be inadvertently overlooked in poorly enhanced images. In glaucoma, as well, assessment of the retinal nerve fiber layer as well as localized thinning of vessels near the optic disc is improved with increased vessel-edge contrast. In hypertensive retinopathy, fine details such as narrowing of arterioles, arteriovenous nicking, as well as hemorrhages, demand high-fidelity structural detail in order to aid in accurate grading. By maintaining vessel continuity as well as highlighting capillary-level structures, OAGF enhances diagnostic conspicuity as well as allows for improved clinical interpretation.”

Comment 9: The results are derived solely from DRIVE and STARE datasets. While they are standard, these datasets lack diversity in pathological variations and image acquisition conditions. Mention this limitation and recommend validation on CHASE_DB1, HRF, or ARIA, which include pathological cases and varied resolutions. Also, consider stating the adaptability of OAGF across different imaging conditions.

Assertion 9: We thank the reviewer for highlighting this important consideration. We admit as well that while widely available and with high-quality expert annotation, the DRIVE as well as STARE datasets suffer significantly in diversity of pathology, image resolution, as well as variability of acquisition conditions. That can reduce generalizability of experimental results. To counter this, we have now mentioned this limitation in text of this revised manuscript in the Conclusion section. We propose further as future research using diverse datasets like CHASE_DB1, HRF, as well as ARIA, with larger variability of pathological condition (e.g., diabetic retinopathy, hypertensive retinopathy) as well as realistic variability of image quality as well as image resolution. We would moreover particularly refer to mention as well that in principle transferable OAGF framework developed here is of gradient-driven, region-aware functional framework formulation, independent of dataset-specific priors as well as of supervised learning. This inherent structural independence is indicative of good generalizing potential of OAGF across imaging conditions, albeit such empirical verification as a future work is needful in this direction.

---

## [Editor Report · Decision Letter 3]

Optimal Anisotropic Guided Filtering in Retinal Fundus Imaging: A Dual Approach to Enhancement and Segmentation

PONE-D-25-11375R3

Dear Dr. Guntreddi,

We’re pleased to inform you that your manuscript has been judged scientifically suitable for publication and will be formally accepted for publication once it meets all outstanding technical requirements.

Kind regards,

Khan Bahadar Khan, Ph.D

Academic Editor

PLOS ONE
---

## [Editor Report · Acceptance letter]

PONE-D-25-11375R3

PLOS ONE

Dear Dr. Guntreddi,

I'm pleased to inform you that your manuscript has been deemed suitable for publication in PLOS ONE. Congratulations! Your manuscript is now being handed over to our production team.

Kind regards,

on behalf of

Dr. Khan Bahadar Khan

Academic Editor

PLOS ONE